# Adaptive Riemannian ADMM for Nonsmooth Optimization: Optimal Complexity without Smoothing

**Kangkang Deng**[*]
College of Science
National University of Defense Technology
Changsha, CHINA
freedeng1208@gmail.com

**Jiachen Jin**[*]
College of Science
National University of Defense Technology
Changsha, CHINA
jinjiachen@nudt.edu.cn

**Jiang Hu**
Yau Mathematical Sciences Center
Tsinghua University
Beijing, CHINA
hujiangopt@gmail.com

**Hongxia Wang**[†]
College of Science
National University of Defense Technology
Changsha, CHINA
wanghongxia@nudt.edu.cn

## Abstract

We study the problem of minimizing the sum of a smooth function and a nonsmooth convex regularizer over a compact Riemannian submanifold embedded in Euclidean space. By introducing an auxiliary splitting variable, we propose an adaptive Riemannian alternating direction method of multipliers (ARADMM), which, for the first time, achieves convergence without requiring smoothing of the nonsmooth term. Our approach involves only one Riemannian gradient evaluation and one proximal update per iteration. Through careful and adaptive coordination of the stepsizes and penalty parameters, we establish an optimal iteration complexity of order $\mathcal{O}(\epsilon^{-3})$ for finding an $\epsilon$-approximate KKT point, matching the complexity of existing smoothing technique-based Riemannian ADMM methods. Extensive numerical experiments on sparse PCA and robust subspace recovery demonstrate that our ARADMM consistently outperforms state-of-the-art Riemannian ADMM variants in convergence speed and solution quality.

## 1 Introduction

Optimization over Riemannian manifolds has garnered significant interest due to its wide-ranging applications in machine learning, statistics, signal processing, and beyond. While the theory and algorithms for smooth manifold optimization have been extensively developed (see [1, 5, 43, 22]), recent years have witnessed a growing need to address nonsmooth objectives, which arise naturally in tasks such as sparse PCA [28], nonnegative PCA [50, 27], and semidefinite programming [7, 45].

---

[*]Equal contribution.
[†]Corresponding author.

39th Conference on Neural Information Processing Systems (NeurIPS 2025).

Formally, we consider the nonsmooth optimization problem on Riemannian manifold

$$\min_{x \in \mathcal{M}} f(x) + h(\mathcal{A}x). \tag{1}$$

where $\mathcal{M}$ is a Riemannian submanifold embedded in $\mathbb{R}^n$, $f : \mathbb{R}^n \to \mathbb{R}$ is a continuously differentiable function, $\mathcal{A} : \mathbb{R}^n \to \mathbb{R}^m$ is a linear mappings, $h : \mathbb{R}^m \to (-\infty, +\infty]$ is a proper closed convex function. Hence, the objective function of (1) can be nonconvex and nonsmooth. By introducing an auxiliary variable $y = \mathcal{A}x$, one obtains the linear constrained problem

$$\min_{x,y} f(x) + h(y), \text{ s.t. } \mathcal{A}x = y, \ x \in \mathcal{M}. \tag{2}$$

To tackle such constrained nonsmooth problems, a natural candidate is the alternating direction method of multipliers (ADMM), which has emerged as a powerful framework for solving large-scale structured optimization problems. One of the earliest attempts to extend ADMM to (2) is the *manifold ADMM* (MADMM) proposed in [31], which adopts the following iteration scheme:

$$\begin{cases} x_{k+1} = \arg \min_{x \in \mathcal{M}} f(x) - \langle \lambda_k, \mathcal{A}x - y_k \rangle + \dfrac{\rho}{2}\|\mathcal{A}x - y_k\|^2, \\ y_{k+1} = \arg \min_{y} h(y) - \langle \lambda_k, \mathcal{A}x_{k+1} - y \rangle + \dfrac{\rho}{2}\|\mathcal{A}x_{k+1} - y\|^2, \\ \lambda_{k+1} = \lambda_k - \rho(\mathcal{A}x_{k+1} - y_{k+1}). \end{cases} \tag{3}$$

where $\lambda$ denotes the Lagrange multiplier and $\rho > 0$ is a penalty parameter. In MADMM, the $x$-subproblem is a smooth Riemannian optimization task that can be solved by using any gradient-based Riemannian algorithm, while the $y$-subproblem admits a closed-form solution via a proximal operator of $h$. Although this algorithm constitutes a direct generalization of the classical Euclidean ADMM, its convergence has remained elusive due to the inherent nonconvexity of Riemannian manifolds. More recently, Li et al. [35] introduced a smoothing technique to reformulate the original nonsmooth problem into a smooth approximation

$$\min_{x \in \mathcal{M}} f(x) + h_\mu(\mathcal{A}x), \tag{4}$$

where $h_\mu$ denotes the smooth approximation of $h$ and $\mu$ is a smoothing parameter. They proposed a smoothed ADMM scheme for solving the surrogate problem (4), and established an $\mathcal{O}(1/\epsilon^4)$ complexity bound. Building upon this idea, [49] further incorporated adaptive smoothing strategies, resulting in an improved algorithm. However, all these methods are designed specifically for smoothed formulations, which rely critically on smoothing parameters, rather than directly addressing the original nonsmooth problem.

In this work, we are interested in the convergence analysis of original Riemannian ADMM (3). A fundamental question arises:

*Can we establish convergence of Riemannian ADMM for the original nonsmooth problem* (1)*, without relying on smoothing?*

This question motivates our study. Our contributions can be summarized as follows:

- We propose a novel adaptive RADMM (ARADMM) for nonsmooth optimization over compact Riemannian submanifolds. Unlike existing smoothing-based Riemannian ADMM methods, our approach achieves convergence without smoothing the nonsmooth regularizer. Moreover, our adaptive strategy dynamically updates the stepsize and penalty parameter during the iterations, avoiding the expensive exact subproblem solutions required by conventional Riemannian ADMM variants. Consequently, each iteration only requires one Riemannian gradient evaluation and one proximal computation, significantly reducing the per-iteration cost. Numerical experiments demonstrate the superior empirical performance of our ARADMM over existing ones.

- Through careful and adaptive coordination of the stepsize and penalty parameter, we establish an optimal iteration complexity of order $\mathcal{O}(\epsilon^{-3})$ for finding an $\epsilon$-approximate KKT point. This matches the best-known complexity achieved by smoothing-based Euclidean and Riemannian ADMM methods, while entirely avoiding the need for smoothing. A key technical innovation is the adaptive selection of dual step sizes and penalty parameters, which explicitly bounds the differences between multipliers by the norms of the corresponding primal iterate differences—an essential property for establishing convergence without requiring exact subproblem solutions.

Table 1: Comparison of the oracle complexity results of several methods in the literature to our method to produce an $\epsilon$-stationary point

| Algorithms | Manifold | Iteration | Without smoothing | Single-loop |
|---|---|---|---|---|
| MADMM [31] | compact | — | Yes | No |
| RADMM [35] | compact | $\mathcal{O}(\epsilon^{-4})$ | No | Yes |
| OADMM [49] | Stiefel | $\mathcal{O}(\epsilon^{-3})$ | No | Yes |
| **this paper** | compact | $\mathcal{O}(\epsilon^{-3})$ | Yes | Yes |

## 1.1 Related works

Most existing works focus on simplified instances of problem (2). These works can be broadly categorized into three classes. First, subgradient and proximal point methods for geodesically convex problems have been studied in [2, 16, 14, 10, 15]. These algorithms typically require stronger assumptions—such as geodesic convexity—and often suffer from slower convergence in practice compared to other approaches. Second, proximal gradient-type methods, such as those in [9, 25, 26], apply in the special case where $\mathcal{A} = \mathcal{I}$. Each iteration of these methods involves solving a subproblem that lacks a closed-form solution, which is typically handled using semismooth Newton techniques. Third, primal-dual methods based on the augmented Lagrangian framework have been developed, including operator splitting algorithms [32], manifold-based augmented Lagrangian methods [12, 55, 11, 47, 48], and Riemannian ADMM variants [31, 35, 49, 53]. Among these, ADMM is particularly attractive due to the separable structure of the objective and constraint in problem (1), which enables efficient and scalable updates.

In the case when the manifold $\mathcal{M}$ is specified by equality constraints $c(x) = 0$, e.g., the Stiefel manifold. Problem (1) can be regarded as a constrained optimization problem with a nonsmooth and nonconvex objective function. Given that $\mathcal{M}$ is often nonconvex, we only list the related works in case of that the constraint functions are nonconvex. Papers [37, 42] propose and study the iteration complexity of augmented Lagrangian methods for solving nonlinearly constrained nonconvex composite optimization problems. The iteration complexity results they achieve for an $\epsilon$-stationary point are both $\tilde{\mathcal{O}}(\epsilon^{-3})$. More specifically, [42] uses the accelerated gradient method of [17] to obtain the approximate stationary point. On the other hand, the authors in [37] obtain such an approximate stationary point by applying an inner accelerated proximal method as in [8, 30], whose generated subproblems are convex. It is worth mentioning that both of these papers make a strong assumption about how the feasibility of an iterate is related to its stationarity. Lin et al. [38] propose an inexact proximal-point penalty method by solving a sequence of penalty subproblems. Under a non-singularity condition, they show a complexity result of $\tilde{\mathcal{O}}(\epsilon^{-3})$. More recently, some works [19, 13, 20] apply ADMM algorithmic framework by penalizing the nonlinear constraint. However, those approaches do not exploit the underlying manifold geometry and essentially reduces to penalty-based methods.

In Table 1, we summarize our complexity results and several existing Riemannian ADMM methods to produce an $\epsilon$-stationary point. We do not list the algorithm in [53] since the subproblem is difficult and requires a strong assumption on the problem: the last block variable cannot appear in the nonsmooth objective. It can easily be shown that our algorithms achieve better oracle complexity results.

## 1.2 Notation

Let $\langle \cdot, \cdot \rangle$ and $\| \cdot \|$ be the Euclidean product and induced Euclidean norm. Given a matrix $A$, we use $\|A\|_F$ to denote the Frobenius norm, $\|A\|_1 := \sum_{ij} |A_{ij}|$ to denote the $\ell_1$ norm. For a vector $x$, we use $\|x\|_2$ and $\|x\|_1$ to denote its Euclidean norm and $\ell_1$ norm, respectively. The distance from $x$ to $\mathcal{C}$ is denoted by $\mathrm{dist}(x, \mathcal{C}) := \min_{y \in \mathcal{C}} \|x - y\|$. We use $\nabla f(x)$ and $\mathrm{grad} f(x)$ to denote the Euclidean gradient and Riemannian gradient of $f$, respectively. $\|\mathcal{A}\|_{op}$ denotes the operator norm of a linear operator $\mathcal{A}$.

## 2 Preliminary

### 2.1 Riemannian optimization

An $n$-dimensional smooth manifold $\mathcal{M}$ is an $n$-dimensional topological manifold equipped with a smooth structure, where each point has a neighborhood that is diffeomorphism to the $n$-dimensional Euclidean space. The tangent space of a manifold $\mathcal{M}$ at $x$ is denoted by $T_x\mathcal{M}$. In this paper, we consider the case that $\mathcal{M}$ is a Riemannian submanifold of an Euclidean space $\mathcal{E}$, the inner product is defined as the Euclidean inner product: $\langle \eta_x, \xi_x \rangle = \mathrm{tr}(\eta_x^\top \xi_x)$. The Riemannian gradient is given by $\mathrm{grad} f(x) = \mathcal{P}_{T_x\mathcal{M}}(\nabla f(x))$, where $\nabla f(x)$ is the Euclidean gradient, $\mathcal{P}_{T_x\mathcal{M}}$ is the projection operator onto the tangent space $T_x\mathcal{M}$. The retraction operator is one of the most important ingredients for manifold optimization, which turns an element of $T_x\mathcal{M}$ into a point in $\mathcal{M}$.

**Definition 2.1** (Retraction, [1]). *A retraction on a manifold $\mathcal{M}$ is a smooth mapping $\mathcal{R} : T\mathcal{M} \to \mathcal{M}$ with the following properties. Let $\mathcal{R}_x : T_x\mathcal{M} \to \mathcal{M}$ be the restriction of $\mathcal{R}$ at $x$. It satisfies*

- $\mathcal{R}_x(0_x) = x$, *where $0_x$ is the zero element of $T_x\mathcal{M}$,*

- $D\mathcal{R}_x(0_x) = id_{T_x\mathcal{M}}$,*where $id_{T_x\mathcal{M}}$ is the identity mapping on $T_x\mathcal{M}$.*

We also give the following definition of vector transport.

**Definition 2.2** (Vector transport, [1]). *Given a Riemannian manifold $\mathcal{M}$, the vector transport $\mathcal{T}_x^y$ is an operator that transports a tangent vector $v \in T_x\mathcal{M}$ to the tangent space $T_y\mathcal{M}$, i.e., $\mathcal{T}_x^y(v) \in T_y\mathcal{M}$. In this paper, we assume $\mathcal{T}_x^y$ is isometric.*

We have the following Lipschitz-type inequalities on the retraction on the compact submanifold.

**Proposition 2.1** ([6, Appendix B]). *Let $\mathcal{R}$ be a retraction operator on a compact submanifold $\mathcal{M}$. Then, there exist two positive constants $\alpha, \beta$ such that for all $x \in \mathcal{M}$ and all $u \in T_x\mathcal{M}$, we have*

$$\|\mathcal{R}_x(u) - x\| \le \alpha\|u\|, \quad \|\mathcal{R}_x(u) - x - u\| \quad \le \beta\|u\|^2. \tag{5}$$

### 2.2 Stationary point and proximal operator

Next we give the definition of $\epsilon$-stationary point for problem (1). Let us first introduce the Lagrangian function $l : \mathcal{M} \times \mathbb{R}^m \times \mathbb{R}^r \to \bar{\mathbb{R}}$ of (1):

$$l(x, y, \lambda) = f(x) + h(y) - \langle \lambda, \mathcal{A}x - y \rangle, \tag{6}$$

where $\lambda$ is the corresponding Lagrangian multiplier. Based on the KKT condition, we give the definition of $\epsilon$-stationary point for (1):

**Definition 2.3.** *We say $x \in \mathcal{M}$ is an $\epsilon$-stationary point of (1) if there exists $y, z \in \mathbb{R}^m$ such that*

$$\begin{cases} \|\mathcal{P}_{T_x\mathcal{M}}(\nabla f(x) - \mathcal{A}^*\lambda)\| \le \epsilon, \\ \quad\quad\quad \mathrm{dist}(-\lambda, \partial h(y)) \le \epsilon, \\ \quad\quad\quad\quad\quad \|\mathcal{A}x - y\| \le \epsilon. \end{cases} \tag{7}$$

*In other words, $(x, y, \lambda)$ is an $\epsilon$-KKT point pair of (1).*

Note that setting $\epsilon = 0$ gives the KKT condition of problem (1).

The following lemma gives the definition of proximal operator for convex function, and the related property.

**Lemma 2.1.** *[3, Lemma 3.3] Let $h$ be a convex function. The proximal operator of $h$ with parameter $\mu > 0$ is given by*

$$prox_{\mu h}(y) = \arg\min_{z \in \mathbb{R}^m} \left\{ h(z) + \frac{1}{2\mu}\|z - y\|^2 \right\}. \tag{8}$$

*Moreover, if $h$ is $\ell_h$-Lipschitz continuous, it holds that*

$$\|x - prox_{\mu h}(x)\| \le \mu\ell_h. \tag{9}$$

## 3 Riemannian ADMM

Throughout this paper, we make the following assumptions.

**Assumption 3.1.** *The following assumptions hold:*

(i) *The manifold $\mathcal{M}$ is a compact Riemannian submanifold embedded in $\mathbb{R}^n$. $f(x)$ and $h(x)$ are both lower bounded, and let $f_* = \inf_x f(x) > -\infty$ and $h_* = \inf_x h(x) > -\infty$.*

(ii) *The function $f$ is $\ell_f$-Lipschitz coninuous and $\ell_{\nabla f}$-smooth on $\mathcal{M}$. The function $h$ is convex and $\ell_h$-Lipschitz continuous.*

(iii) *The linear mapping $\mathcal{A}$ satisfies $\|\mathcal{A}\|_{op} \leq \sigma_A$.*

Assumption 3.1 (i) includes many common manifolds, such as sphere, Stiefel manifold and Oblique manifold, etc. This implies $\mathcal{M}$ is a bounded and closed set, i.e., there exists a finite constant $\mathcal{D}$ such that $\mathcal{D} = \max_{x,y \in \mathcal{M}} \|x - y\|$. Assumption 3.1 (ii) implies that for any $x, y \in \mathcal{M}$, it holds that

$$\|\nabla f(x) - \nabla f(y)\| \leq \ell_{\nabla f} \|x - y\|. \tag{10}$$

### 3.1 Challenges in bounding dual updates in Riemannian ADMM

The convergence analysis of ADMM algorithms for nonconvex problems typically relies on establishing a sufficient descent property of the augmented Lagrangian (AL) function, which serves as a potential function. However, since ADMM belongs to the class of primal-dual algorithms, the update of the dual variables introduces an ascent term, commonly expressed as $\|\lambda_{k+1} - \lambda_k\|$. Controlling this ascent term is critical for ensuring convergence.

In the Euclidean setting, this term is usually bounded via the optimality condition of the subproblem associated with the primal variable, see, for example, [46, 18, 21, 4]. Consider the iterative scheme in (3). When the manifold $\mathcal{M} = \mathbb{R}^n$, the optimality condition for the $x$-subproblem at iteration $k$ reads

$$\nabla f(x_{k+1}) + \rho \mathcal{A}^*(\mathcal{A}x_{k+1} - y_k) - \mathcal{A}^*\lambda_k = 0.$$

Taking differences of the optimality conditions at iterations $k$ and $k+1$, one can derive a bound on $\|\lambda_{k+1} - \lambda_k\|$ by the difference of corresponding primal iterates using the Lipschitz continuity of $\nabla f$. However, in the Riemannian setting, the corresponding optimality condition involves projections onto tangent spaces:

$$\mathcal{P}_{T_{x_{k+1}}\mathcal{M}} \left(\nabla f(x_{k+1}) + \rho \mathcal{A}^*(\mathcal{A}x_{k+1} - y_k) - \mathcal{A}^*\lambda_k\right) = 0.$$

Due to the nonlinear geometry of the manifold, the tangent space changes at each iteration, making it impossible to directly apply the difference technique used in the Euclidean case. This leads to significant challenges in bounding the difference between dual variables, as the projection operators prevent the necessary alignment of optimality conditions across iterations.

A common workaround is to smooth the nonsmooth term associated with the $y$-variable. By replacing the nonsmooth regularizer $h$ with a smooth approximation $h_\mu$, one can leverage the Lipschitz continuity of the gradient $\nabla h_\mu$ to bound the difference of multipliers. Specifically, the optimality condition for the smoothed $y$-subproblem becomes

$$0 = \nabla h_\mu(y_{k+1}) + \rho(\mathcal{A}x_k - y_{k+1}) - \lambda_k.$$

Thus the desired bound on the multipliers can be obtained by the Lipschitz continuity of $\nabla h_\mu$. Please refer to [35, 49] for more details. However, this approach fundamentally changes the problem structure from nonsmooth optimization to smooth optimization, and various gradient-based methods can be used instead of ADMM. For instance, [41] developed a Riemannian homotopy smoothing algorithm based on this idea.

In contrast, our approach works directly with the original, nonsmoothed ADMM framework, without introducing any smoothing to the problem. To overcome the challenge posed by the changing tangent spaces and nonsmooth regularizer, we introduce an adaptive strategy for selecting the dual stepsizes and penalty parameters. Specifically, we introduce an adaptive penalty parameter $\rho_k$ and replace the

**Algorithm 1** Adaptive Riemannian ADMM for solving (1).

---

**Input**: initial point $x_0, y_0, \lambda_0, \rho_0, \gamma_0$, parameters $c_\rho, c_\gamma$.

1: **for** $k = 0, \cdots, K - 1$ **do**
2:     Update auxiliary variable $y_{k+1}$ via

$$y_{k+1} = \arg\min_{y \in \mathbb{R}^d} \mathcal{L}_{\rho_k}(x_k, y, \lambda_k). \tag{14}$$

3:     Denote $\Phi_k(x) := \mathcal{L}_{\rho_k}(x, y_{k+1}, \lambda_k)$ and obtain $x_{k+1}$ by single gradient step:

$$x_{k+1} = \mathcal{R}_{x_k}(-\tau_k \mathrm{grad}\Phi_k(x_k)). \tag{15}$$

4:     Update the dual step size $\gamma_{k+1}$ via

$$\gamma_{k+1} = \min\left(\frac{\gamma_0 \|\mathcal{A}x_0 - y_0\| \log^2 2}{\|\mathcal{A}x_{k+1} - y_{k+1}\|(k+1)^2 \log(k+2)}, \frac{c_\gamma}{k^{1/3} \log^2(k+1)}\right). \tag{16}$$

5:     Update the dual variable $\lambda_{k+1}$ via

$$\lambda_{k+1} = \lambda_k - \gamma_{k+1}(\mathcal{A}x_{k+1} - y_{k+1}). \tag{17}$$

6: **end for**
**Output**: $(x_K, y_K, \lambda_K)$.

---

original dual stepsize with $\gamma_k$ in original Riemannian ADMM (3). Leveraging the properties of the Moreau envelope, we then obtain

$$\lambda_{k+1} - \lambda_k = \gamma_{k+1}(\mathcal{A}x_{k+1} - y_{k+1})$$
$$= \gamma_{k+1}\left(\mathcal{A}x_k - \frac{\lambda_k}{\rho_k} - \mathrm{prox}_{h/\rho_k}\left(\mathcal{A}x_k - \frac{\lambda_k}{\rho_k}\right)\right) + \gamma_{k+1}\mathcal{A}(x_{k+1} - x_k) + \gamma_{k+1}\frac{\lambda_k}{\rho_k}.$$

The first term can be bounded using properties of the proximal operator and the Moreau envelope, as shown in (9). To further control $\|\lambda_k\|$, we design the dual stepsize $\gamma_{k+1}$ adaptively, ensuring that the multiplier difference is bounded by the primal iterate difference. More details are referred to Algorithm 1 and Lemma 3.1.

### 3.2 Adaptive Riemannian ADMM

We construct the corresponding augmented Lagrangian function:

$$\mathcal{L}_\rho(x, y, \lambda) := f(x) + h(y) - \langle \lambda, \mathcal{A}x - y \rangle + \frac{\rho}{2}\|\mathcal{A}x - y\|^2. \tag{11}$$

Algorithm 1 details the iterative process. For the update rule of the dual variable $\lambda_{k+1}$, we use a different sequence $\{\gamma_k\}$ to replace original sequence $\{\rho_k\}$, and result in the following update:

$$\lambda_{k+1} = \lambda_k - \gamma_{k+1}(\mathcal{A}x_{k+1} - y_{k+1}). \tag{12}$$

Here, we refer to $\gamma_{k+1}$ as the dual step size, which is updated by the following form:

$$\gamma_{k+1} = \min\left(\frac{\gamma_0 \|\mathcal{A}x_0 - y_0\| \log^2 2}{\|\mathcal{A}x_{k+1} - y_{k+1}\|(k+1)^2 \log(k+2)}, \frac{c_\gamma}{k^{1/3} \log^2(k+1)}\right). \tag{13}$$

Our algorithm alternately updates the variables in the order of $(y, x, \lambda)$, which follows from [23]. The increasing sequence of penalty parameters $\rho_k$ and the dual update are responsible for continuously enforcing the constraints. The particular choice of dual step sizes $\gamma_k$ in Algorithm 1 ensures that the difference between the dual variables $\lambda_k$ and $\lambda_{k+1}$ remains bounded, which is crucial for the convergence analysis.

Thanks to the careful choice of the penalty parameter and step size, we establish the following key lemma, which ensures that the difference between successive dual variables, $\|\lambda_{k+1} - \lambda_k\|$, can be effectively controlled.

**Lemma 3.1.** *Let Assumptions 3.1 hold. Suppose the sequence $\{x_k, y_k, \lambda_k\}_{k=1}^K$ is generated by the Algorithm 1. The following inequality holds*

$$\|\lambda_{k+1}\| \leq \frac{\gamma_0 \pi^2}{6} \|\mathcal{A}x_0 - y_0\| = \lambda_{\max}. \tag{18}$$

*Moreover we have that*

$$\|\lambda_{k+1} - \lambda_k\| \leq \frac{\gamma_{k+1}}{\rho_k}(\ell_h + \lambda_{\max}) + \gamma_{k+1}\sigma_A \|x_{k+1} - x_k\|. \tag{19}$$

*Proof of Lemma 3.1.* We first show that $\lambda_k$ is bounded. By step 5 in Algorithm 1, we have

$$\|\lambda_{k+1}\| \leq \sum_{l=1}^{k+1} \gamma_l \|\mathcal{A}x_l - y_l\| \leq \sum_{l=1}^{\infty} \gamma_l \|\mathcal{A}x_l - y_l\|$$

$$\leq \gamma_0 \|\mathcal{A}x_0 - y_0\| \log^2 2 \sum_{l=1}^{\infty} \frac{1}{(l+1)^2 \log(l+2)} \leq \frac{\gamma_0 \pi^2}{6} \|\mathcal{A}x_0 - y_0\| = \lambda_{\max}, \tag{20}$$

where the last inequality utilize that $\log(l+2) > 1$ for $l \geq 1$ and $\sum_{l=1}^{\infty} \frac{1}{(l+1)^2} = \frac{\pi^2}{6}$. Again using step 5 in Algorithm 1, one can bounds the difference between $\lambda_{k+1}$ and $\lambda_k$:

$$\|\lambda_{k+1} - \lambda_k\| = \gamma_{k+1}\|\mathcal{A}x_{k+1} - y_{k+1}\|$$

$$= \gamma_{k+1}\|\mathcal{A}x_k - y_{k+1} - \frac{\lambda_k}{\rho_k}\| + \gamma_{k+1}\|\mathcal{A}\|_{op}\|x_{k+1} - x_k\| + \gamma_{k+1}\|\frac{\lambda_k}{\rho_k}\|$$

$$\leq \gamma_{k+1}\|(\mathcal{A}x_k - \frac{\lambda_k}{\rho_k} - \text{prox}_{\frac{h}{\rho_k}}(\mathcal{A}x_k - \frac{\lambda_k}{\rho_k}))\| + \gamma_{k+1}\sigma_A\|x_{k+1} - x_k\| + \gamma_{k+1}\|\frac{\lambda_k}{\rho_k}\|$$

$$\leq \frac{\gamma_{k+1}}{\rho_k}(\ell_h + \lambda_{\max}) + \gamma_{k+1}\sigma_A\|x_{k+1} - x_k\|, \tag{21}$$

where the first inequality uses $\|\mathcal{A}\|_{op} \leq \sigma_A$ from Assumption 3.1 and the update rule of $y_{k+1}$ in (14), the second inequality follows from (9) and (20). The proof is completed.

$\square$

Now we provide the main convergence result of our algorithm ARADMM. In particular, we show that under certain assumptions, the ARADMM can achieve a oracle complexity of $\mathcal{O}(\epsilon^{-3})$.

**Theorem 3.1.** *Suppose that Assumptions 3.1 hold. Let the sequence $\{x_k, y_k, \lambda_k\}_{k=1}^K$ be generated by Algorithm 1. Let us denote $\bar{\lambda}_k = \lambda_{k-1} - \rho_{k-1}(\mathcal{A}x_k - y_k)$, $\rho_k = c_\rho k^{1/3}$ and $\tau_k = c_\tau k^{-1/3}$, where $c_\tau, c_\rho$ satisfy*

$$c_\tau \leq \min\{\frac{1}{C}, \frac{1}{M}, \frac{1}{16c_\gamma \alpha^2 \sigma_A^2}\}, \quad c_\rho \geq 1, \tag{22}$$

*where $C := (\alpha L_p + \zeta)G + \alpha(\ell_{\nabla f} + \sigma_A^2)$, $M := \alpha^2(\ell_{\nabla f} + \sigma_A^2) + 2G\beta$, and $G, \alpha, \beta$ are given in (5) and Lemma A.2, $L_p, \zeta$ are defined in Lemma A.1. Then for any given positive integer $K > 2$, there exists $\kappa \in [\lceil K/2 \rceil, K]$ such that*

$$\|\mathcal{P}_{T_{x_\kappa}\mathcal{M}}(-\mathcal{A}^*\bar{\lambda}_\kappa) + \text{grad}f(x_\kappa)\| \leq \frac{8\sqrt{\mathcal{G}}}{\sqrt{c_\tau}}(K+1)^{-1/3},$$

$$dist(-\bar{\lambda}_\kappa, \partial h(y_\kappa)) \leq \frac{4\sigma_A \alpha \sqrt{\mathcal{G}}}{\sqrt{c_\tau}}(K+1)^{-1/3},$$

$$\|\mathcal{A}x_\kappa - y_\kappa\| \leq 8\sigma_A \alpha \sqrt{c_\tau \mathcal{G}}(K-2)^{-2/3} + \frac{2(\ell_h + \lambda_{\max})}{c_\rho}(K-2)^{-1/3}. \tag{23}$$

*where $\mathcal{G}$ is a constant given in the proof.*

Theorem 3.1 establishes that, given $\epsilon > 0$, our algorithm achieves an iteration complexity of $\mathcal{O}(\epsilon^{-3})$. Since Algorithm 1 is a single-loop method that requires only one gradient evaluation of $f$, one

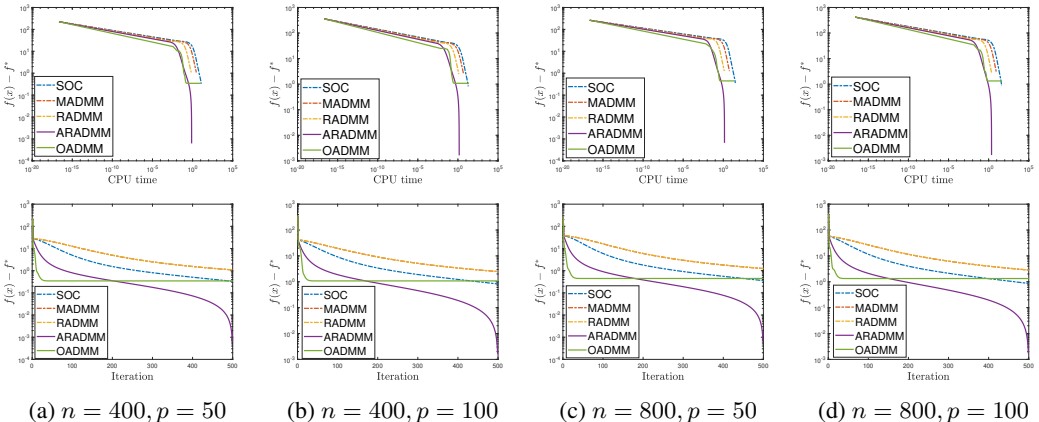

| (a) $n = 400, p = 50$ | (b) $n = 400, p = 100$ | (c) $n = 800, p = 50$ | (d) $n = 800, p = 100$ |

Figure 1: Comparison with ADMM-type methods for solving (24) with different $(n, p)$, $m = n$ and $\mu = 0.01$.

computation of the proximal operator of $h$ and retraction operator per iteration—both of which are computationally inexpensive—its overall operation complexity is of the same order as its iteration complexity.

We would like to clarify that this condition in (22) is not essential. For example, since $\tau_k = c_\tau k^{-1/3}$, the condition on $c_\tau$ is essentially imposed to ensure that $\tau_k$ satisfies the required bound. This condition can always be satisfied after a sufficiently large number of iterations without requiring $c_\tau$. That is, for any $c_\tau$, there exists an integer $k_0 > 0$ such that for all $k > k_0$, $\tau_k$ satisfies the condition.

## 4 Applications and Numerical Experiments

In this section, we investigate the numerical performance of the proposed algorithm and report comparative results with existing methods. All experiments are performed in MATLAB R2023b on a 64-bit laptop equipped with Intel i9-13900HX CPU and 32.0 GB RAM. We denote the final objective values as "obj" and report the CPU time in seconds. All the results are averaged across 10 repeated experiments with random initializations.

### 4.1 Sparse Principal Component Analysis

Sparse principal component analysis [28, 29] is a cornerstone technique for high-dimensional data analysis, identifying principal components with sparse loadings. Given a data matrix $A \in \mathbb{R}^{m \times n}$, the problem of recovering the top $p$ ($p < \min\{m, n\}$) sparse loading vectors is formulated as:

$$\min_X F(X) := -\frac{1}{2}\text{Tr}(X^\top A^\top A X) + \mu\|X\|_1, \text{ s.t. } X \in \text{St}(n, p), \tag{24}$$

where $\mu > 0$ is a regularization parameter, $\text{Tr}(X)$ denotes the trace of matrix $X$ and $\text{St}(n, p) = \{X \in \mathbb{R}^{n \times p} : X^\top X = I_p\}$ is the Stiefel manifold.

We evaluate our proposed Algorithm 1 to solve (24) and compare it with four ADMM-type methods: SOC [33], MADMM [31], RADMM [35] and OADMM [49]. For ARADMM, we set $\gamma_0 = c_\gamma = 50$, $\rho_0 = 5$, $c_\rho = 1$ and $c_\tau = 0.2$. For OADMM, we set $\xi = 0.1$ and other parameters are the same as their originals. For the other three algorithms, we follow the same settings as [35], where the parameters are optimized through grid searches. All algorithms use identical random initializations and terminate when $|F(X_{k+1}) - F(X_k)| \leq 10^{-8}$ or after 500 iterations. The data matrix $A \in \mathbb{R}^{m \times n}$ is generated randomly and the entries follow the standard Gaussian distribution. Figure 1 shows the objective value versus iterations and CPU time, where $f^*$ is the minimum value across all methods. ARADMM achieves significantly lower objective values and converges faster than other ADMM.

We also compare ARADMM with the Riemannian subgradient method (RSG) [14, 36] and the accelerated manifold proximal gradient method (AManPG) [24]. We set the step size $\eta_k \equiv 0.005$

Table 2: Comparison with RSG and AManPG for solving (24) with $\mu = 0.01$.

| Settings | RSG | | AManPG | | ARADMM | |
|---|---|---|---|---|---|---|
| $(n, m, p)$ | Obj | CPU | Obj | CPU | Obj | CPU |
| (300, 20, 8) | -3.9517 | 0.7970 | -4.2671 | 0.8555 | **-4.4071** | **0.4264** |
| (400, 30, 10) | -5.5595 | 1.1105 | -5.8129 | 1.0796 | **-6.0378** | **0.5383** |
| (500, 40, 12) | -7.1168 | 1.9584 | -7.3048 | 1.3820 | **-7.6569** | **0.6888** |
| (600, 50, 14) | -7.2259 | 2.5911 | -7.4608 | 1.7482 | **-7.9029** | **0.8701** |

Table 3: Comparison with ALM-type methods for solving (24). Here "Spa" denotes the sparsity level, defined as the proportion of entries with a magnitude less than $10^{-4}$.

| Settings | ALMSSN | | | ALMSRTR | | | ARADMM | | |
|---|---|---|---|---|---|---|---|---|---|
| $(n, m, p, \mu)$ | Obj | CPU | Spa | Obj | CPU | Spa | Obj | CPU | Spa |
| (1500, 20, 8, 0.5) | 6.0212 | 1.6349 | 93.10 | 4.0169 | 1.2679 | 98.38 | **3.9552** | 0.9417 | **99.93** |
| (2000, 40, 10, 0.6) | 8.9390 | 2.7063 | 94.03 | 5.9553 | 1.9281 | 99.85 | **5.9190** | 1.3272 | **99.95** |
| (2500, 60, 12, 0.8) | **9.4527** | 2.9387 | 99.61 | 9.5296 | 3.6644 | 99.95 | 9.4747 | 1.8394 | **99.96** |
| (3000, 80, 15, 1) | 15.7085 | 4.7114 | 94.65 | **14.8418** | 4.4791 | 99.96 | 14.8470 | 2.2669 | **99.97** |

for RSG, while the code of AManPG is provided by [25]. For ARADMM, we set $\gamma_0 = c_\gamma = 10^3$, $\rho_0 = 10^2$, $c_\rho = 1$ and $c_\tau = 10^{-4}$. The termination rules are the KKT conditions with an accuracy tolerance of $10^{-8}$. Table 2 shows that ARADMM consistently achieves lower objective values at a significantly faster rate than RSG and AManPG, demonstrating both efficiency and solution quality.

Finally, we compare with two ALM-type algorithms: ALMSSN [55] and ALMSRTR [51], which use a semismooth Newton method and a Riemannian trust region method, respectively, to solve the augmented Lagrangian subproblem on manifolds. The codes for these algorithms are obtained from related work. The parameters for ARADMM and the termination rules follow the same settings as in above experiments. From Table 3, we can see that ARADMM generates sparse solutions faster than ALMSSN and ALMSRTR, especially when $\mu$ is large.

## 4.2 Regularized Linear Classifier Over Sphere Manifold

Consider a classification task involving training pairs $\{a_i, b_i\}_{i=1}^N$, where $a_i \in \mathbb{R}^m$ and $b_i \in \{-1, 1\}$ for all $i \in [N]$. The objective is to estimate a linear classifier parameter $x$ on the sphere manifold $\mathrm{S}^{m-1} := \{x \in \mathbb{R}^m : x^\top x = 1\}$ that minimizes a smooth nonconvex loss [54, 52] with $\ell_1$-regularization:

$$\min_{x \in \mathrm{S}^{m-1}} \sum_{i=1}^N \left(1 - \frac{1}{1 + \exp(-b_i x^\top a_i)}\right)^2 + \mu\|x\|_1, \tag{25}$$

For data generation, the true parameter $x$ is sampled from $\mathcal{N}(0, I_m)$. and projected onto $\mathcal{S}^{m-1}$. The features $\{a_i\}_{i=1}^N$ are sampled independently and the labels $b_i$ are set to 1 if $x^\top a_i + \epsilon_i > 0$, where noise $\epsilon_i \sim \mathcal{N}(0, \sigma^2)$, and -1, otherwise. All algorithms use the identical random initialization and terminate when $|F(X_{k+1}) - F(X_k)| \leq 10^{-8}$ or after 500 iterations, with $\mu = 0.2$ fixed in (25).

We set $\gamma_0 = \rho_0 = c_\gamma = 100$, $c_\rho = 1$ and $c_\tau = 0.05$ for ARADMM, and use the same settings as in the SPCA experiments for OADMM. For the other methods, we set penalty parameter $\rho = 150$ and step size $\eta = 0.01$. From Table 4, we can see that ARADMM and MADMM quickly decrease the objective value, whereas both ARADMM and SOC achieve a lower objective value of the outputs. Moreover, ARADMM is more advantageous than existing methods in more challenging scenarios. In short, ARADMM is more efficient in terms of the CPU time and objective value for test instances.

## 4.3 Robust Subspace Recovery and Dual Principal Component Pursuit

Robust subspace recovery (RSR) [34, 40] addresses the challenge of fitting a linear subspace to data corrupted by outliers. Given a data set $Y = [\mathcal{X}, \mathcal{O}]\Gamma \in \mathbb{R}^{n \times (p_1 + p_2)}$, where columns of $X \in \mathbb{R}^{n \times p_1}$ span a $d$-dimensional inlier subspace $\mathcal{S}$, columns of $\mathcal{O} \in \mathbb{R}^{n \times p_2}$ represent outliers without a linear

Table 4: Comparison with ADMM-type methods for solving (25).

| Settings | SOC | | MADMM | | RADMM | | OADMM | | ARADMM | |
|---|---|---|---|---|---|---|---|---|---|---|
| $(m, n, \sigma^2)$ | Obj | CPU | Obj | CPU | Obj | CPU | Obj | CPU | Obj | CPU |
| (200, 1000, 1) | **0.7004** | 1.1009 | 0.7340 | 0.0729 | 0.7340 | 0.0876 | 0.7282 | 0.1024 | 0.7370 | **0.0507** |
| (400, 5000, 5) | 0.6877 | 2.2876 | 0.8267 | 0.1701 | 0.8267 | 0.1864 | 0.7288 | 0.2799 | **0.6875** | **0.1073** |
| (600, 10000, 10) | 0.6469 | 7.8231 | 0.9216 | 0.6692 | 0.9216 | 0.7153 | 0.6665 | 1.0664 | **0.6464** | **0.4197** |
| (800, 20000, 50) | 0.6606 | 30.7367 | 1.0398 | 2.2673 | 1.0396 | 2.7062 | 0.6871 | 2.7062 | **0.6602** | **1.4167** |

Table 5: Comparison with ADMM-type methods for solving (27).

| Settings | | SOC | | MADMM | | RADMM | | ARADMM | |
|---|---|---|---|---|---|---|---|---|---|
| $p$ | $(n, p_1, p_2)$ | obj | CPU | obj | CPU | obj | CPU | obj | CPU |
| 4 | (30, 100, 500) | 286.5284 | 1.4835 | 286.4820 | 0.7677 | 286.4599 | 0.0486 | **286.3336** | **0.0057** |
| | (40, 125, 750) | 363.2060 | 2.6384 | 363.1336 | 1.4823 | 363.0977 | 0.0136 | **362.8826** | **0.0119** |
| | (50, 150, 1000) | 423.6242 | 2.7054 | 423.5769 | 2.3196 | 423.5352 | 0.0378 | **423.2783** | **0.0136** |
| 6 | (30, 100, 500) | 431.2076 | 2.4063 | 431.1518 | 1.4842 | 431.1275 | 0.0160 | **431.0405** | **0.0089** |
| | (40, 125, 750) | 542.7145 | 2.8961 | 542.6425 | 0.7155 | 542.5957 | 0.0540 | **542.4900** | **0.0112** |
| | (50, 150, 1000) | 637.8058 | 2.6239 | 637.7484 | 0.7584 | 637.6906 | 0.0972 | **637.3598** | **0.0176** |

structure, and $\Gamma \in \mathbb{R}^{(p_1+p_2) \times (p_1+p_2)}$ is an unknown permutation matrix, the goal is to recover $\mathcal{S}$ or cluster the points into inliers and outliers. Dual principal component pursuit (DPCP) [44, 56] is a recently proposed approach to RSR that seeks a hyperplane containing all inliers via the nonconvex nonsmooth optimization:

$$\min_{x \in \mathbb{R}^n} \|Y^\top x\|_1, \text{ s.t. } \|x\|_2 = 1. \tag{26}$$

Here $Y \in \mathbb{R}^{n \times p}$ is a given matrix. In [44, 56] it is shown that solving (26) yields a vector orthogonal to $\mathcal{S}$, provided that outliers $p_2$ is at most of the order of $\mathcal{O}(p_1^2)$. For known $d$, one can recover $\mathcal{S}$ as the intersection of the $p := n - d$ orthogonal hyperplanes containing $\mathcal{X}$, which amounts to solve the following matrix optimization problem:

$$\min_{X \in \mathbb{R}^{n \times p}} F(X) := \|Y^\top X\|_1, \text{ s.t. } X^\top X = I_p. \tag{27}$$

We focus on the DPCP formulation (27) of the RSR problem, and compare our ARADMM with SOC [33], MADMM [31] and RADMM [35]. For ARADMM, we set $c_\rho = 1$, $\gamma_0 = 700$, $\rho_0 = 5$, $c_\tau = 10^{-2}$ and $c_\gamma = 0.6$. The codes of other methods are provided by [35], where we set the stepsize $\eta = 10^{-5}$. All methods terminate when $|F(X_{k+1}) - F(X_k)| \le 10^{-6}$ or after 5000 iterations. Table 5 shows that, for all cases, ARADMM consistently achieves lower objective values and very shorter computation times than SOC, MADMM and RADMM. This efficiency is due to: ARADMM has a cheap per-iteration complexity, where all steps have closed-form solutions; the adaptive penalty parameter $\rho_k$ and the dual step size $\gamma_k$ dynamically balance the enforcement of constraints with the convergence of the algorithm in an effective manner. More numerical results see Appendix C.

## 5 Conclusion

Our work introduces an adaptive Riemannian ADMM (ARADMM) that, for the first time, solves composite optimization on compact manifolds with a nonsmooth regularizer without any smoothing, while requiring only one Riemannian gradient evaluation and one proximal update per iteration. By dynamically tuning both stepsizes and penalty parameters—and carefully relating dual increments to primal changes—we prove that ARADMM attains the optimal $\mathcal{O}(\epsilon^{-3})$ iteration complexity for finding an $\epsilon$-approximate KKT point, matching the best-known guarantees of smoothing-based methods at a far lower per-iteration cost. Extensive experiments on sparse PCA and robust subspace recovery confirm that our adaptive scheme converges faster and yields higher-quality solutions than existing Riemannian ADMM variants.

**Limitations:** While ARADMM demonstrates strong theoretical guarantees and practical performance, our current analysis is limited to a general nonconvex nonsmooth setting without leveraging specific structural properties such as the Kurdyka–Łojasiewicz (KL) inequality, which could potentially yield sharper convergence rates.

## Acknowledgments

We sincerely thank four anonymous reviewers for their valuable and constructive feedback, which has greatly improved our work. This work was supported by the following grants: the National Key Research and Development Program of China (No. 2020YFA0713504), the National Natural Science Foundation of China (Grant No. 12471401, 12401419).

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

# A    Useful lemmas

The following lemma shows that the Lipschitz continuity of $\mathrm{grad} f(x)$ can be deduced by the Lipschitz continuity of $\nabla f(x)$.

**Lemma A.1.** *Suppose that $\mathcal{M}$ is a compact submanifold embedded in the Euclidean space, given $x, y \in \mathcal{M}$ and $u \in T_x \mathcal{M}$, the vector transport $\mathcal{T}$ is defined as $\mathcal{T}_x^y(u) := D\mathcal{R}_x[\xi](u)$, where $y = \mathcal{R}_x(\xi)$. Denote $\zeta := \max_{x \in conv(\mathcal{M})} \|D^2 \mathcal{R}_x(\cdot)\|$ and $G := \max_{x \in \mathcal{M}} \|\nabla f(x)\|$. Let $L_p$ be the Lipschitz constant of $\mathcal{P}_{T_x \mathcal{M}}$ over $x \in \mathcal{M}$ in sense that for any $x, y \in \mathcal{M}, \xi \in \mathbb{R}^n$,*

$$\|\mathcal{P}_{T_x \mathcal{M}}(\xi) - \mathcal{P}_{T_y \mathcal{M}}(\xi)\| \le L_p \|\xi\| \|x - y\|. \tag{28}$$

*For a function $f$ with Lipschitz continuous gradient with constant L, i.e., $\|\nabla f(x) - \nabla f(y)\| \le L\|x - y\|$, then we have that*

$$\|\mathrm{grad} f(x) - \mathcal{T}_y^x(\mathrm{grad} f(y))\| \le ((\alpha L_p + \zeta)G + \alpha L)\|\xi\|. \tag{29}$$

Next, we provide the definition of retraction smoothness. This concept plays a crucial role in the convergence analysis of the algorithm proposed in the following section.

**Definition A.1.** *[6, Retraction smooth] A function $f : \mathcal{M} \to \mathbb{R}$ is said to be retraction smooth (short to retr-smooth) with constant $\ell$ and a retraction $\mathcal{R}$, if for $\forall\, x, y \in \mathcal{M}$ it holds that*

$$f(y) \le f(x) + \langle \mathrm{grad} f(x), \eta \rangle + \frac{\ell}{2} \|\eta\|^2, \tag{30}$$

*where $\eta \in T_x \mathcal{M}$ and $y = \mathcal{R}_x(\eta)$.*

We next demonstrate that $\Phi_k(x)$ is retr-smooth with some constant $L_k$ by Definition A.1. The Euclidean gradient $\nabla \Phi_k(x)$ has the following form:

$$\nabla \Phi_k(x) = \nabla f(x) + \rho_k \mathcal{A}^* (\mathcal{A}x - y_{k+1} - \lambda_k / \rho_k). \tag{31}$$

The following lemma provides the essential insight.

**Lemma A.2.** *Suppose that Assumption 3.1 holds. If $c_\rho \ge 1$, there exists finite M such that $\Phi_k$ is retr-smooth in the sense that*

$$\Phi_k(\mathcal{R}_x(\eta)) \le \Phi_k(x) + \langle \eta, \mathrm{grad} \Phi_k(x) \rangle + \frac{M \rho_k}{2} \|\eta\|^2 \tag{32}$$

*for all $\eta \in T_x \mathcal{M}$, where $M := \alpha^2(\ell_{\nabla f} + \sigma_A^2) + 2G\beta$.*

*Proof.* By Assumption 3.1, one can easily shows that $\nabla \Phi_k$ is Lipschitz continuous with constant $\ell_{\nabla f} + \rho_k \sigma_A^2$. Since $\nabla \psi_k$ is continuous on the compact manifold $\mathcal{M}$, there exists $G > 0$ such that $\|\nabla \psi_k(x)\| \le G$ for all $x \in \mathcal{M}$ and any $k > 0$. Then the proof is completed by combining (5) with Lemma 2.7 in [6]. $\square$

# B  Proof of Section 3.2

We now give the following descent lemma in term of augmented Lagrangian function.

**Lemma B.1.** *Suppose that Assumptions 3.1 hold. Let the sequence $\{x_k, y_k, \lambda_k\}_{k=1}^K$ be generated by Algorithm 1. Let $\rho_k = c_\rho k^{1/3}$ and $\tau_k = c_\tau k^{-1/3}$, where $c_\tau$ satisfies*

$$c_\tau \leq \frac{1}{M}, c_\rho \geq 1 \tag{33}$$

*where $M$ is defined in Lemma 3.1 and $G, \alpha, \beta$ are defined in Lemma A.2. Then*

$$\mathcal{L}_{\rho_{k+1}}(x_{k+1}, y_{k+1}, \lambda_{k+1}) - \mathcal{L}_{\rho_k}(x_k, y_k, \lambda_k)$$
$$\leq \left( \frac{1}{\gamma_{k+1}} + \frac{c_\rho^3}{3} \frac{1}{\gamma_{k+1}^2 \rho_k^2} \right) \|\lambda_{k+1} - \lambda_k\|^2 - \frac{\tau_k}{2} \|\mathrm{grad}_x \mathcal{L}_{\rho_k}(x_k, y_{k+1}, \lambda_k)\|^2. \tag{34}$$

*Proof.* By the step 2 in Algorithm 1, we have

$$\mathcal{L}_{\rho_k}(x_k, y_{k+1}, \lambda_k) \leq \mathcal{L}_{\rho_k}(x_k, y_k, \lambda_k). \tag{35}$$

It follows from (15) and Lemma A.2 that

$$\Phi_k(x_{k+1}) \leq \Phi_k(x_k) + \langle \mathrm{grad}\Phi_k(x_k), -\tau_k \mathrm{grad}\Phi_k(x_k) \rangle + \frac{M\rho_k \tau_k^2}{2} \|\mathrm{grad}\Phi_k(x_k)\|^2$$
$$\leq \Phi_k(x_k) - \frac{\tau_k}{2} \|\mathrm{grad}\Phi_k(x_k)\|^2, \tag{36}$$

where the second inequality uses $\tau_k \leq \frac{1}{M\rho_k}$. This is due to the fact that $\rho_k = c_\rho k^{1/3}$ and $c_\tau$ satisfies (22). Therefore, we have from the definition of $\Phi_k$ in Algorithm 1 that

$$\mathcal{L}_{\rho_k}(x_{k+1}, y_{k+1}, \lambda_k) \leq \mathcal{L}_{\rho_k}(x_k, y_{k+1}, \lambda_k) - \frac{\tau_k}{2} \|\mathrm{grad}_x \mathcal{L}_{\rho_k}(x_k, y_{k+1}, \lambda_k)\|^2 \tag{37}$$

For the dual variable, it follows from (12) that

$$\mathcal{L}_{\rho_k}(x_{k+1}, y_{k+1}, \lambda_{k+1}) - \mathcal{L}_{\rho_k}(x_{k+1}, y_{k+1}, \lambda_k) = \frac{1}{\gamma_{k+1}} \|\lambda_{k+1} - \lambda_k\|^2. \tag{38}$$

Combining (35), (37) and (38), we have

$$\mathcal{L}_{\rho_{k+1}}(x_{k+1}, y_{k+1}, \lambda_{k+1}) - \mathcal{L}_{\rho_k}(x_k, y_k, \lambda_k)$$
$$\leq \mathcal{L}_{\rho_{k+1}}(x_{k+1}, y_{k+1}, \lambda_{k+1}) - \mathcal{L}_{\rho_k}(x_{k+1}, y_{k+1}, \lambda_{k+1}) + \mathcal{L}_{\rho_k}(x_{k+1}, y_{k+1}, \lambda_{k+1}) - \mathcal{L}_{\rho_k}(x_k, y_k, \lambda_k)$$
$$\leq \frac{\rho_{k+1} - \rho_k}{2} \|\mathcal{A}x_{k+1} - y_{k+1}\|^2 + \frac{1}{\gamma_{k+1}} \|\lambda_{k+1} - \lambda_k\|^2 - \frac{\tau_k}{2} \|\mathrm{grad}_x \mathcal{L}_{\rho_k}(x_k, y_{k+1}, \lambda_k)\|^2$$
$$\leq \frac{\rho_{k+1} - \rho_k}{2\gamma_{k+1}^2} \|\lambda_{k+1} - \lambda_k\|^2 + \frac{1}{\gamma_{k+1}} \|\lambda_{k+1} - \lambda_k\|^2 - \frac{\tau_k}{2} \|\mathrm{grad}_x \mathcal{L}_{\rho_k}(x_k, y_{k+1}, \lambda_k)\|^2, \tag{39}$$

where the last inequality uses (12). Now we bound the first term. Consider $p(x) = x^{1/3}$, it follows from the first order characterization that $p(x+1) \leq p(x) + p'(x) = x^{1/3} + \frac{1}{3}x^{-2/3}$. Then we have that

$$\rho_{k+1} - \rho_k = c_\rho((k+1)^{1/3} - k^{1/3}) \leq \frac{c_\rho}{3} k^{-2/3} = \frac{c_\rho^3}{3} \frac{1}{\rho_k^2}.$$

Plugging this into (39) completes the proof. $\qquad\square$

**Lemma B.2.** *Suppose that Assumptions 3.1 hold. Let the sequence $\{x_k, y_k, \lambda_k\}_{k=1}^K$ be generated by Algorithm 1. Let us denote $\bar{\lambda}_k = \lambda_{k-1} - \rho_{k-1}(\mathcal{A}x_k - y_k)$, $\rho_k = c_\rho k^{1/3}$ and $\tau_k = c_\tau k^{-1/3}$, where $c_\tau$ and $c_\rho$ satisfies*

$$c_\tau \leq \min\{\frac{1}{C}, \frac{1}{M}\}, c_\rho \geq 1. \tag{40}$$

*where $C$ is defined in the proof and $M$ is given in Lemma A.2. Then*

$$\|\mathcal{P}_{T_{x_k}\mathcal{M}}(-\mathcal{A}^* \bar{\lambda}_k) + \mathrm{grad}f(x_k)\| \leq 2\|\mathrm{grad}\mathcal{L}_{\rho_{k-1}}(x_{k-1}, y_k, \lambda_{k-1})\|, \tag{41}$$

$$\mathrm{dist}(-\bar{\lambda}_k, \partial h(y_k)) \leq \sigma_A \alpha \|\mathrm{grad}\mathcal{L}_{\rho_{k-1}}(x_{k-1}, y_k, \lambda_{k-1})\|, \tag{42}$$

$$\|\mathcal{A}x_k - y_k\| \leq \frac{\ell_h + \lambda_{\max}}{\rho_{k-1}} + \sigma_A \alpha \tau_{k-1} \|\mathrm{grad}\mathcal{L}_{\rho_{k-1}}(x_{k-1}, y_k, \lambda_{k-1})\|. \tag{43}$$

*Proof of Lemma B.2.* It follows from the formulas of $\bar{\lambda}_k$ and $\mathcal{L}_{\rho_{k-1}}(x_k, y_k, \lambda_{k-1})$ that

$$\begin{aligned}
\mathrm{grad}_x \mathcal{L}_{\rho_{k-1}}(x_k, y_k, \lambda_{k-1}) &= \mathcal{P}_{T_{x_k}\mathcal{M}}\left(\nabla f(x_k) + \rho_{k-1}\mathcal{A}^*(\mathcal{A}x_k - y_k - \frac{\lambda_{k-1}}{\rho_{k-1}})\right) \\
&= \mathcal{P}_{T_{x_k}\mathcal{M}}\left(\nabla f(x_k) - \mathcal{A}^*\bar{\lambda}_k\right).
\end{aligned} \tag{44}$$

Since that $\nabla_x \mathcal{L}_{\rho_{k-1}}(x_k, y_k, \lambda_{k-1})$ is Lipscitz continuous with constant $\ell_{\nabla f} + \rho_{k-1}\sigma_A^2$, it follows from Lemma A.1 that $\mathrm{grad}_x \mathcal{L}_{\rho_{k-1}}(x_{k-1}, y_k, \lambda_{k-1})$ is Lipschitz continuous with

$$\ell_k = (\alpha L_p + \zeta)G + \alpha(\ell_{\nabla f} + \rho_{k-1}\sigma_A^2) \le C\rho_{k-1},$$

where $C := (\alpha L_p + \zeta)G + \alpha(\ell_{\nabla f} + \sigma_A^2)$, which is due to $\rho_{k-1} \ge 1$.

$$\begin{aligned}
&\|\mathcal{P}_{T_{x_k}\mathcal{M}}\left(\nabla f(x_k) - \mathcal{A}^*\bar{\lambda}_k\right)\| = \|\mathrm{grad}_x \mathcal{L}_{\rho_{k-1}}(x_k, y_k, \lambda_{k-1})\| \\
\le& \|\mathcal{T}_{x_{k-1}}^{x_k}\mathrm{grad}_x \mathcal{L}_{\rho_{k-1}}(x_{k-1}, y_k, \lambda_{k-1})\| + \|\mathrm{grad}_x \mathcal{L}_{\rho_{k-1}}(x_k, y_k, \lambda_{k-1}) - \mathcal{T}_{x_{k-1}}^{x_k}\mathrm{grad}_x \mathcal{L}_{\rho_{k-1}}(x_{k-1}, y_k, \lambda_{k-1})\| \\
\le& \|\mathrm{grad}_x \mathcal{L}_{\rho_{k-1}}(x_{k-1}, y_k, \lambda_{k-1})\| + C\rho_{k-1}\|\tau_{k-1}\mathrm{grad}_x \mathcal{L}_{\rho_{k-1}}(x_{k-1}, y_k, \lambda_{k-1})\| \\
\le& 2\|\mathrm{grad}_x \mathcal{L}_{\rho_{k-1}}(x_{k-1}, y_k, \lambda_{k-1})\|,
\end{aligned}$$

where the second inequality follows from Lemma A.1 and the fact that $\mathcal{T}$ is isometric, the last inequality uses $\tau_{k-1} \le \frac{1}{C\rho_{k-1}}$ by the condition for $c_\tau$. It follows from (14) that

$$0 \in \lambda_{k-1} - \rho_{k-1}(\mathcal{A}x_{k-1} - y_k) + \partial h(y_k).$$

By the definition of $\bar{\lambda}_k$, we have that

$$\begin{aligned}
\mathrm{dist}(-\bar{\lambda}_k, \partial h(y_k)) &= \rho_{k-1}\sigma_A\|x_k - x_{k-1}\| \\
&\le \sigma_A \alpha \rho_{k-1}\tau_{k-1}\|\mathrm{grad}_x \mathcal{L}_{\rho_{k-1}}(x_{k-1}, y_k, \lambda_{k-1})\| \\
&\le \sigma_A \alpha \|\mathrm{grad}_x \mathcal{L}_{\rho_{k-1}}(x_{k-1}, y_k, \lambda_{k-1})\|,
\end{aligned}$$

where the first inequality uses (5), the second inequality follows $\tau_k \le 1/L_k \le 1/\rho_k$, which is implied by (36). For (43), it follows that

$$\begin{aligned}
\|\mathcal{A}x_k - y_k\| &= \frac{1}{\gamma_k}\|\lambda_k - \lambda_{k-1}\| \\
&\le \frac{\ell_h + \lambda_{\max}}{\rho_{k-1}} + \sigma_A\|x_k - x_{k-1}\| \\
&\le \frac{\ell_h + \lambda_{\max}}{\rho_{k-1}} + \sigma_A \alpha \tau_{k-1}\|\mathrm{grad}_x \mathcal{L}_{\rho_{k-1}}(x_{k-1}, y_k, \lambda_{k-1})\|,
\end{aligned}$$

where the first inequality uses (19), the second inequality follows (5). The proof is completed.

$\square$

**Theorem B.1.** *Suppose that Assumptions 3.1 hold. Let the sequence $\{x_k, y_k, \lambda_k\}_{k=1}^K$ be generated by Algorithm 1. Let $\rho_k = c_\rho k^{1/3}$ and $\tau_k = c_\tau k^{-1/3}$, where $c_\tau, c_\rho$ satisfy*

$$c_\tau \le \min\{\frac{1}{C}, \frac{1}{M}, \frac{1}{16c_\gamma \alpha^2 \sigma_A^2}\}, \quad c_\rho \ge 1, \tag{45}$$

*where $G, \alpha, \beta$ are defined in Lemma A.2. There exists an constant $\mathcal{G}$ such that*

$$\sum_{k=1}^K \frac{\tau_k}{4}\|grad\mathcal{L}_{\rho_k}(x_k, y_{k+1}, \lambda_k)\|^2 \le \mathcal{G}.$$

*Proof.* Combining with Lemmas 3.1 and B.1 yields

$$\mathcal{L}_{\rho_{k+1}}(x_{k+1}, y_{k+1}, \lambda_{k+1}) - \mathcal{L}_{\rho_k}(x_k, y_k, \lambda_k) \le -\frac{\tau_k}{2}\|\mathrm{grad}_x \mathcal{L}_{\rho_k}(x_k, y_{k+1}, \lambda_k)\|^2$$

$$+ \left(\frac{1}{\gamma_{k+1}} + \frac{c_\rho^3}{3}\frac{1}{\gamma_{k+1}^2 \rho_k^2}\right)\left(\frac{2\gamma_{k+1}^2}{\rho_k^2}(\ell_h + \lambda_{\max})^2 + 2\gamma_{k+1}^2 \sigma_A^2 \|x_{k+1} - x_k\|^2\right)$$

$$\le -\frac{\tau_k}{2}\|\mathrm{grad}_x \mathcal{L}_{\rho_k}(x_k, y_{k+1}, \lambda_k)\|^2 + \left(\frac{2\gamma_{k+1}}{\rho_k^2} + \frac{2c_\rho^3}{3\rho_k^4}\right)(\ell_h + \lambda_{\max})^2 + \left(2\gamma_{k+1}\sigma_A^2 + \frac{2c_\rho^3 \sigma_A^2}{\rho_k^2}\right)\|x_{k+1} - x_k\|^2$$

$$\le -\frac{\tau_k}{2}\|\mathrm{grad}_x \mathcal{L}_{\rho_k}(x_k, y_{k+1}, \lambda_k)\|^2 + \left(\frac{2c_\gamma}{c_\rho}\frac{1}{k\log^2(k+2)} + \frac{2}{3c_\rho}k^{-4/3}\right)(\ell_h + \lambda_{\max})^2$$

$$+ 4\gamma_{k+1}\tau_k^2 \alpha^2 \sigma_A^2 \|\mathrm{grad}_x \mathcal{L}_{\rho_k}(x_k, y_{k+1}, \lambda_k)\|^2$$

$$\le -\frac{\tau_k}{4}\|\mathrm{grad}_x \mathcal{L}_{\rho_k}(x_k, y_{k+1}, \lambda_k)\|^2 + \left(\frac{2c_\gamma}{c_\rho}\frac{1}{k\log^2(k+2)} + \frac{2}{3c_\rho}k^{-4/3}\right)(\ell_h + \lambda_{\max})^2$$

where the third inequality follows from (5) and step 3 in Algorithm 1, and

$$\frac{2\gamma_{k+1}}{\rho_k^2} + \frac{2c_\rho^3}{3\rho_k^4} \le \frac{2c_\gamma}{c_\rho}\frac{1}{k\log^2(k+2)} + \frac{2}{3c_\rho}k^{-4/3},$$

the last inequality follows $\tau_k \le c_\tau \le \frac{1}{16c_\gamma Q^2 \sigma_A^2}$. Summing over $k = 1$ to $K$:

$$\sum_{k=1}^{K}\frac{\tau_k}{4}\|\mathrm{grad}\mathcal{L}_{\rho_k}(x_k, y_{k+1}, \lambda_k)\|^2 \le \underbrace{\mathcal{L}_{\rho_1}(x_1, y_1, \lambda_1) - \mathcal{L}_{\rho_{K+1}}(x_{K+1}, y_{K+1}, \lambda_{K+1})}_{a}$$

$$+ \frac{2c_\gamma(\ell_h + \lambda_{\max})^2}{c_\rho}\underbrace{\sum_{k=1}^{K}\frac{1}{k\log^2(k+2)}}_{b} + \frac{2(\ell_h + \lambda_{\max})^2}{3c_\rho}\underbrace{\sum_{k=1}^{K}k^{-4/3}}_{c}$$

Now we bound $a, b, c$, respectively. For $a$, let us denote $\phi_k := \mathcal{L}_{\rho_k}(x_k, y_k, \lambda_k)$, then we have

$$\begin{aligned}\phi_k &= f(x_k) + h(y_k) - \langle\lambda_k, \mathcal{A}x_k - y_k\rangle + \frac{\rho_k}{2}\|\mathcal{A}x_k - y_k\|^2 \\ &\ge f(x_k) + h(y_k) - \|\lambda_k\|\|\mathcal{A}x_k - y_k\| \\ &\ge f_* + h_* - \lambda_{\max}\left(\frac{\ell_h + \lambda_{\max}}{\rho_{k-1}} + \sigma_A\|x_k - x_{k-1}\|\right) \\ &\ge f_* + h_* - \lambda_{\max}(\ell_h + \lambda_{\max} + \sigma_A \mathcal{D}).\end{aligned}\tag{46}$$

We let $\mathcal{C}_1 := f_* + h_* - \lambda_{\max}(\ell_h + \lambda_{\max} + \sigma_A \mathcal{D})$, and obtain $a \le \phi_1 - \mathcal{C}_1$. For $b$, following [42], there exists a constant $\mathcal{C}_2$ such that

$$b \le \sum_{k=1}^{\infty}\frac{1}{k\log^2(k+2)} \le \mathcal{C}_2.\tag{47}$$

For $c$, since $4/3 > 1$, the sequence $\sum_{k=1}^{\infty} k^{-4/3}$ is convergence, there exists a constant $\mathcal{C}_3$ such that $c \le \mathcal{C}_3$. Combining with those terms, one concludes that

$$\sum_{k=1}^{K}\frac{\tau_k}{4}\|\mathrm{grad}\mathcal{L}_{\rho_k}(x_k, y_{k+1}, \lambda_k)\|^2 \le \mathcal{C}_1 + \left(\frac{2c_\gamma}{c_\rho}\mathcal{C}_2 + \frac{2}{3c_\rho}\mathcal{C}_3\right)(\ell_h + \lambda_{\max})^2.\tag{48}$$

Let us denote $\mathcal{G} := \mathcal{C}_1 + \left(\frac{2c_\gamma}{c_\rho}\mathcal{C}_2 + \frac{2}{3c_\rho}\mathcal{C}_3\right)(\ell_h + \lambda_{\max})^2$. The proof is completed.

$\square$

*Proof of Theorem 3.1.* It follows from Theorem B.1 that

$$\begin{aligned}\min_{\lceil K/2\rceil \le k \le K}\|\mathrm{grad}\mathcal{L}_{\rho_{k-1}}(x_{k-1}, y_k, \lambda_{k-1})\|^2 &\left(\sum_{k=\lceil K/2\rceil}^{K}\tau_k\right) \\ \le \sum_{k=\lceil K/2\rceil}^{K}\tau_k\|\mathrm{grad}\mathcal{L}_{\rho_{k-1}}(x_{k-1}, y_k, \lambda_{k-1})\|^2 &\le 4\mathcal{G},\end{aligned}\tag{49}$$

which implies that

$$\min_{\lceil K/2 \rceil \le k \le K} \|\text{grad}\mathcal{L}_{\rho_{k-1}}(x_{k-1}, y_k, \lambda_{k-1})\|^2 \le \frac{4\mathcal{G}}{\sum_{k=\lceil K/2 \rceil}^K \tau_k}. \tag{50}$$

Now we bound $\sum_{k=\lceil K/2 \rceil}^K \tau_k$. Since $\tau_k = c_\tau k^{-1/3}$ and the function $x^{-1/3}$ is monotonically decreasing, one has that

$$
\begin{aligned}
\sum_{k=\lceil K/2 \rceil}^K \tau_k &\ge c_\tau \sum_{k=\lceil K/2 \rceil}^K \int_k^{k+1} x^{-1/3}\, \mathrm{d}x \\
&= c_\tau \sum_{k=\lceil K/2 \rceil}^K \frac{3}{2}\left[(k+1)^{2/3} - k^{2/3}\right] \\
&= \frac{3c_\tau}{2}\left[(K+1)^{2/3} - (K/2+1)^{2/3}\right] \\
&\ge \frac{3c_\tau}{2}\left[\frac{2}{3}(K+1)^{-1/3}((K+1) - (K/2+1))\right] \\
&= \frac{c_\tau}{2}K(K+1)^{-2/3} \ge \frac{c_\tau}{4}(K+1)^{2/3}.
\end{aligned}
\tag{51}
$$

where the second inequality uses concavity of $x^{2/3}$. Together with (49), there exists $\kappa \in [\lceil K/2 \rceil, K]$ such that

$$\|\text{grad}\mathcal{L}_{\rho_{\kappa-1}}(x_{\kappa-1}, y_\kappa, \lambda_{\kappa-1})\|^2 \le \frac{16\mathcal{G}}{c_\tau}(K+1)^{-2/3}. \tag{52}$$

Using (52) and Lemma B.2, it is easily shown that

$$
\begin{aligned}
\|\mathcal{P}_{T_{x_\kappa}\mathcal{M}}(-\mathcal{A}^*\bar{\lambda}_\kappa) + \text{grad}f(x_\kappa)\| &\le \frac{8\sqrt{\mathcal{G}}}{\sqrt{c_\tau}}(K+1)^{-1/3}, \\
\text{dist}(-\bar{\lambda}_\kappa, \partial h(y_\kappa)) &\le \frac{4\sigma_A\alpha\sqrt{\mathcal{G}}}{\sqrt{c_\tau}}(K+1)^{-1/3}.
\end{aligned}
\tag{53}
$$

which proves the first two term in (23). For the last term, we have that

$$
\begin{aligned}
\|\mathcal{A}x_\kappa - y_\kappa\| &\le \sigma_A\alpha\tau_{\kappa-1}\|\text{grad}\mathcal{L}_{\rho_{\kappa-1}}(x_{\kappa-1}, y_\kappa, \lambda_{\kappa-1})\| + \frac{\ell_h + \lambda_{\max}}{\rho_{\kappa-1}} \\
&\le 8\sigma_A\alpha\sqrt{c_\tau\mathcal{G}}(K-2)^{-1/3}(K+1)^{-1/3} + \frac{2(\ell_h + \lambda_{\max})}{c_\rho}(K-2)^{-1/3} \\
&\le 8\sigma_A\alpha\sqrt{c_\tau\mathcal{G}}(K-2)^{-2/3} + \frac{2(\ell_h + \lambda_{\max})}{c_\rho}(K-2)^{-1/3},
\end{aligned}
\tag{54}
$$

where we use $\tau_\kappa = c_\tau(\kappa-1)^{-1/3} \le c_\tau(\frac{K-2}{2})^{-1/3} \le 2c_\tau(K-2)^{-1/3}$ and $\rho_{\kappa-1} = c_\rho(\kappa-1)^{1/3} \ge c_\rho(\frac{K-2}{2})^{1/3} \ge \frac{c_\rho}{2}(K-2)^{1/3}$. Combining with (53) completes the proof. $\qquad\square$

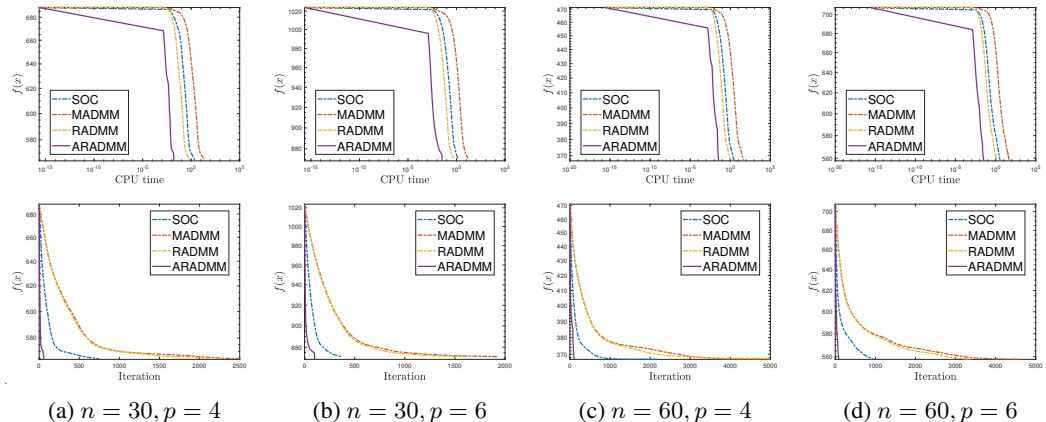

(a) $n = 30, p = 4$      (b) $n = 30, p = 6$      (c) $n = 60, p = 4$      (d) $n = 60, p = 6$

Figure 2: Comparison with ADMM-type methods for solving (27) with different $(n, p)$ and $(p_1, p_2) = (150, 1000)$.

## C    More numerical results

This section emphasizes that the condition in (22) is not essential. Here, we consider the condition $c_\rho \geq 1$ and focus on the DPCP formulation (27), comparing our ARADMM method with SOC [33], MADMM [31] and RADMM [35]. For ARADMM, we reset $\gamma_0 = 10^3$, $c_\rho = 0.6$ and retain $\rho_0, c_\tau, c_\gamma$. The parameter settings for the baseline algorithms follow [35]. We report the average results over 10 trials with random initializations in Tables 6 and 7. The function values for the sequence on the manifold versus CPU time (in seconds) and iteration number are recorded in Figures 2 and 3. It is shown that the results remain consistent with those reported using $c_\rho = 1$ in subsection 4.3.

From a theoretical point of view, the condition appears initially in Lemma A.2, where it was used to simplify the expression in the following inequality:

$$\Phi_k(\mathcal{R}_x(\eta)) \leq \Phi_k(x) + \langle \eta, \mathrm{grad}\Phi_k(x) \rangle + \frac{\alpha^2(\ell_{\nabla f} + \sigma_A^2) + 2G\beta}{2} \|\eta\|^2.$$

Assuming $c_\rho \geq 1$ implies $\rho_k \geq 1$, which simplifies the bound to:

$$\alpha^2(\ell_{\nabla f} + \sigma_A^2) + 2G\beta \leq (\alpha^2(\ell_{\nabla f} + \sigma_A^2) + 2G\beta)\rho_k.$$

However, this assumption is not strictly necessary. In fact, we can relax it by identifying a threshold index $\tilde{k} > 0$, such that for all $k \geq \tilde{k}$, we have $\rho_k \geq 1$. This does not affect the final convergence guarantees—the same convergence rate still holds. This type of strategy has been employed in prior work (e.g., [39]) to mitigate the reliance on precise problem constants. This is why $c_\rho = 0.6$ also works well empirically here. In fact, the condition for other parameters can also be eliminated by this strategy.

Table 6: Numerical results of ADMM-type methods for solving (27) with different $(n, p)$ and $(p_1, p_2) = (150, 1000)$.

| Settings | SOC | | MADMM | | RADMM | | ARADMM | |
|---|---|---|---|---|---|---|---|---|
| $(n, p)$ | Obj | CPU | Obj | CPU | Obj | CPU | Obj | CPU |
| $(30, 4)$ | 568.6116 | 6.8864 | 568.4316 | 18.0771 | 569.3725 | 0.7801 | **568.3537** | **0.0244** |
| $(30, 6)$ | 852.8768 | 2.0746 | 852.8180 | 9.3013 | 852.8157 | 0.4351 | **852.8125** | **0.0191** |
| $(60, 4)$ | 366.1369 | 2.6194 | 365.8585 | 16.2287 | 365.9267 | 0.8130 | **365.6622** | **0.0245** |
| $(60, 6)$ | 553.5725 | 3.6961 | 553.5749 | 43.4826 | 554.0265 | 1.0868 | **553.5650** | **0.1541** |

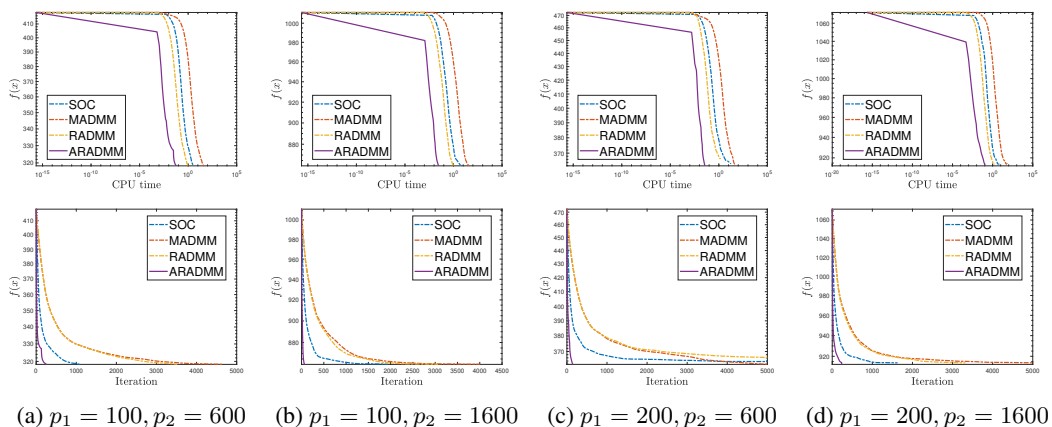

(a) $p_1 = 100, p_2 = 600$  (b) $p_1 = 100, p_2 = 1600$  (c) $p_1 = 200, p_2 = 600$  (d) $p_1 = 200, p_2 = 1600$

Figure 3: Comparison with ADMM-type methods for solving (27) with different $(p_1, p_2)$ and $(n, p) = (45, 5)$.

Table 7: Numerical results of ADMM-type methods for solving (27) with different $(p_1, p_2)$ and $(n, p) = (45, 5)$.

| Settings | SOC | | MADMM | | RADMM | | ARADMM | |
|---|---|---|---|---|---|---|---|---|
| $(p_1, p_2)$ | obj | CPU | obj | CPU | obj | CPU | obj | CPU |
| $(100, 600)$ | 317.0240 | 1.9779 | 316.9943 | 29.2210 | 317.3680 | 0.8800 | **316.8880** | **0.0238** |
| $(100, 1600)$ | 859.3896 | 3.5202 | 859.3150 | 26.3167 | 859.3417 | 0.2115 | **859.0431** | **0.0206** |
| $(200, 600)$ | 356.6751 | 2.5096 | 356.5580 | 30.6229 | 357.4611 | 0.8100 | **356.4748** | **0.0273** |
| $(200, 1600)$ | 908.4715 | 5.2544 | 908.0242 | 12.7438 | 909.4992 | 0.7122 | **907.8516** | **0.1086** |

