# OpenReview forum: "Adaptive Riemannian ADMM for Nonsmooth Optimization: Optimal Complexity without Smoothing"
_NeurIPS.cc/2025/Conference — NeurIPS 2025 poster_

### Official Review · Reviewer_LuUY · 2025-06-19

**Clarity:** 3
**Significance:** 3
**Originality:** 3
**Rating:** 5
**Confidence:** 4

**Summary:**

This paper addresses the problem of minimizing the sum of a smooth function and a non-smooth convex regularizer over a compact Riemannian submanifold embedded in Euclidean space. The authors propose an adaptive Riemannian alternating direction method of multipliers (ARADMM), which achieves convergence for the first time without requiring the smoothing of the non-smooth term. In contrast to conventional Riemannian ADMM methods that require solving a subproblem exactly at each iteration, the proposed approach involves only one Riemannian gradient evaluation and one proximal update per iteration. A sophisticated dual step size design ensures the convergence of the algorithm. Finally, extensive numerical experiments demonstrate the superiority of ARADMM in terms of both convergence speed and solution quality.

**Questions:**

- Regarding the dual step size $\beta_k$, could you clarify the inspiration behind this choice? Is there any similar approach discussed in previous literature? Is there a similar step size design in the Euclidean case? Additionally, what is the role of the constant $\log^2 2$ in $\beta_k$? It seems that its critical function is not evident in the proof. A clearer explanation of this would help highlight the originality and significance of your work.
- The choice of $c_\tau$ appears to be a delicate adjustment in practice, especially since the Lipschitz constant of the function may not be easily known. In your experiments, did you test different values of $c_\tau$? How do the results vary with different selections of $c_\tau$? Clarifying this aspect would further demonstrate the practical applicability of your algorithm.
- I noticed that in the Sparse PCA experiments, you compared your method with ManPG. As far as I know, the method AManPG in reference [19] achieves faster and better results for Sparse PCA. Why was this method not included in your comparison? Additionally, it seems that comparisons with ALM-based algorithms could also be relevant, such as those in [41] and [Zhang, C. et al.]$^1$. Furthermore, the experimental dimensions appear to be limited to less than 1000. Have you conducted experiments with larger dimensions? Providing an explanation for the exclusion of these comparisons, or additional experiments, would help us better understand the advantages of your proposed algorithm.
- In the proof section, is there a complete proof or a reference for Lemma A.1? Also, the condition $c_\rho \geq 1$ seems to be incorrect and should be $c_\rho \leq 1$, which matches the value used in your experiments. Please carefully review the corresponding locations and the relationship between $\tau_k$ and $\rho_k$. Furthermore, equation (29) seems to be missing a $\lambda_0$, and the summation symbol in the second inequality should have the index $l=0$ instead of $l=1$. In line 421, the first equation should be an inequality, and in line 418, $\ell_k$ should be replaced with $\ell_{k-1}$.

Reference
1. Zhang, C., Xiao, R., Huang, W., & Jiang, R. (2024). Riemannian Trust Region Methods for SC 1 Minimization. Journal of Scientific Computing, 101(2), 32.

**Ethical Concerns:**

["NO or VERY MINOR ethics concerns only"]

**Final Justification:**

The authors have addressed the raised concerns and obtained solid results; the additional experiments provided make me feel confident about the practical feasibility of the proposed algorithm. Therefore, I am willing to raise my score by one point.

**Limitations:**

Yes.

**Paper Formatting Concerns:**

None.

**Quality:**

3

**Strengths And Weaknesses:**

### Strengths
The structure of the paper is well-organized, and the logical flow is clear and coherent. The authors present, for the first time, an algorithm that achieves the convergence complexity of smoothing-based methods without the need for smoothing techniques. The numerical experiments demonstrate impressive results, further validating the proposed method. Additionally, the clever dual step size design provides a guarantee for controlling the boundedness of the Lagrangian multipliers, which seems to be an innovative contribution.

### Weaknesses
While the overall logic of the paper is clear, there are some areas that could be improved:
- Some equation numbers, such as those in equations (11) and (13), appear to be unnecessary. These redundant labels impact readability.
- In line 60, the discussion on “the best-known complexity achieved by smoothing-based Euclidean and Riemannian ADMM methods” could benefit from the addition of references to support the claim.
- The usage of $\tilde{\mathcal{O}}$ should be explained for readers who are less familiar with complexity notation.
- In line 172, the formula should be $\lambda_k - \lambda_{k+1}$, and the final inequality should be an equality.
- The equality in equation (18) should be defined by the symbol used in the paper.
- In line 405, the definitions of various parameters should reference their initial definitions, and in line 431, the symbol $D$ should refer back to its previous definition for clarity.
- The paper would benefit from reducing typographical errors. There are some typographical errors:
    - In equations (1) and (3), the periods should be replaced with commas.
    - In line 24, "a linear mappings" should be corrected to "a linear mapping."
    - In Proposition 2.1, the second inequality has an extra space.
    - In line 430, $Q$ should be $\alpha$.
    - In line 439, "Corollary" should be changed to "Theorem."

---

> ### Author Rebuttal · Authors · 2025-07-29
>
> Thank you for your positive feedback on the structure and clarity of the paper.
>
> **Reply to Weakness**
> * In the revised version, we will remove such redundant labels to improve readability and streamline the presentation.
> * The mention of “Euclidean” in line 60 was a typo — we intended to refer only to smoothing-based Riemannian ADMM methods. We have corrected this in the revised manuscript and added the reference [38].
> * We will include a brief explanation of the $\tilde{\mathcal{O}}$ notation in the revised manuscript. Specifically, we will clarify that $\tilde{\mathcal{O}}(\cdot)$ hides logarithmic factors in the standard $\mathcal{O}(\cdot)$ notation.
> * We have revised the formula in line 172 to correctly reflect the expression $\lambda_k - \lambda\_{k+1}$, and changed the final inequality to an equality as appropriate.
> * We have revised the equation to explicitly define the symbol used in the paper:
> $$
> \\quad \\quad \\|\lambda\_{k+1}\\| \leq  \lambda\_{\max},
> $$
> where $\lambda\_{\max}: = \frac{\beta_0 \pi^2 }{6} \\|\mathcal{A}x_0 - y_0\\|$.
> * In the revised manuscript, we have clarified the definitions of all relevant parameters:
> The parameter $M$ is defined in Theorem 3.1. The parameters $\alpha$ and $\beta$ are defined in Equation (5).  The constant $G$ is defined in Lemma A.2. The symbol $\mathcal{D}$ in line 431 now explicitly refers back to its definition in line 142, where we define $
>   \mathcal{D} = \max\_{x, y \in \mathcal{M}} \\|x - y\\|.
>   $
> We have revised the manuscript to ensure that all such symbols are clearly defined and consistently referenced.
> * We have corrected all of them in the revised manuscript. We have also thoroughly proofread the entire manuscript to eliminate other potential typographical errors.
>
> **Reply to Questions**
> * **Regrading the parameter $\beta_k$.** We thank the reviewer for the thoughtful question about the design of the dual step size $\beta_k$. The choice of $\beta_k$ is composed of two parts:
>
>    - **The first part** follows a design in \[10] (refer to manuscript). Its original purpose is to ensure that the dual variable in the augmented Lagrangian method remains bounded, i.e., $\\|\lambda_k\\|$ is uniformly controlled. We incorporate this term, along with the penalty parameter and the dual stepsize, to derive a sharp bound on $\\|\lambda_{k+1} - \lambda_k\\|$, as shown in Lemma 3.1. The constant $\log^2 2$ included in this part does not play a critical role in the analysis—it is included solely to maintain consistency with the $\log(k+1)$ term in the denominator.
>
>    - **The second part** is introduced to achieve optimal iteration complexity. This term arises from the analysis in the proof of Theorem 3.1, where it is chosen to balance the convergence terms optimally. In particular, the denominator $\log^2(k)$ is introduced to **avoid the appearance of a $\log K$** factor in the final convergence rate — as reflected in Equation (45) of our paper.
> We will provide a more detailed explanation of the motivation and design of $\beta_k$ in the revised manuscript to clarify both its theoretical and practical implications.
> * **Regarding the parameter $c_\{\tau}$**
>   - We acknowledge that the choice of $c_\{\tau}$ may depend on certain problem-specific constants, such as the Lipschitz constant. However, we would like to clarify that this dependence is **not essential**. Since $\tau_k = c\_\tau k^{-1/3}$, the condition on $c_\tau$ is essentially imposed to ensure that $\tau_k$ satisfies the required bound, i.e.,  $\tau_k < \min\\{  \frac{1}{C}, \frac{1}{M}, \frac{1}{16 c\_{\beta}\alpha^2 \sigma_A^2}\\}$ (Theorem 3.1). This condition can always be satisfied after a sufficiently large number of iterations without requiring $c_\tau$. That is, for any $c_\tau$, there exists an integer $k_0 > 0$ such that for all $k > k_0$,  $\tau_k$ satisfies the condition.  Importantly, this does not affect the final convergence guarantees—the same convergence rate still holds.
>   - In our experiments, we tested several different values of $c_\{\tau}$ and selected the one that yielded the best performance. While the results do exhibit some variation across different choices of $c_\{\tau}$, we observed that the algorithm is not particularly sensitive to this parameter. We will clarify this point after the main theorem and provide additional details in the experimental section.
> * **Regarding the experiments**.
> While preparing additional experiments as suggested, we identified an implementation error in one of the experiments reported in Table 2 (comparison with ManPG in our manuscript): due to an inadvertent use of an output from another method as input to our algorithm, the reported runtime was significantly underestimated. We emphasize that this issue occurred only in this particular experiment; all other experiments were correctly implemented and remain unaffected.
> We have corrected the implementation and re-run this specific experiment (see **Table 1** below). As suggested by the reviewer, we have replaced ManPG with a stronger baseline, AManPG, in this experiment. The updated results still confirm the competitive performance of our method, and our main findings remain unchanged. All other experiments and theoretical results are unaffected by this issue.
> We sincerely apologize for the oversight and appreciate the reviewer’s comments, which helped us catch and resolve this error promptly.
>   - **Comparison with AmanPG**. We compare ARADMM with the Riemannian subgradient method (RSG) and the accelerated manifold proximal gradient method (AManPG) \[1]. ARADMM uses the same settings as in our original experiment. We set the step size $\eta_k \equiv 10^{-2}$ for RSG, while the code of AManPG is provided by [1]. **Table 1** reports the averaged results across 10 repeated experiments with random initializations. The termination rules are the KKT conditions with an accuracy tolerance of $10^{-8}$.
>
>   - **Comparison with ALM-based methods**. We compare two ALM-based algorithms: ALMSSN \[2] and ALMSRTR \[3], which use a semismooth Newton method and a Riemannian trust region method, respectively, to solve the augmented Lagrangian subproblem on manifolds. The codes for these algorithms are obtained from related work.  **Table 2** reports the function value (denoted by “Obj”), CPU time (in seconds), and the percentage of zero entries in the output (denoted by “Spa”), where the sparsity level is defined as the proportion of entries with a magnitude less than $10^{-4}$, and the results are averaged across 10 repeated experiments with random initializations. ARADMM uses the same settings as in our SPCA experiment. The termination rules are the KKT conditions with an accuracy tolerance of $10^{-8}$.
>
> * **Regrading the choice of $c\_{\rho}$**
>   -  We have re-run the experiments with $c\_{\rho} = 1$. The result is shown in **Table 1** (See reply to 3rd Reviewer **xPLN** due to word limit). It is shown that the results remain consistent with those reported using $c\_{\rho} = 0.6$.
>   - In our response to 3nd Reviewer **xPLN**, we realized that the condition on $c_\tau$ was missing its dependence on $c_\rho$. We have now corrected it to:
> $$
> c\_{\tau} \leq \frac{1}{c_{\rho}} \min\\{  \frac{1}{C}, \frac{1}{M}, \frac{1}{16 c\_{\beta}\alpha^2 \sigma_A^2}\\}.
> $$
>    - We have carefully addressed and corrected all the typographical errors. We sincerely thank the reviewer for the detailed and thoughtful comments, which have significantly improved the clarity and accuracy of our manuscript.
>
> ---
>
>  **Table 1: Comparison of RSG, AManPG, and ARADMM for solving the SPCA problem with $\mu = 0.01$**
>
> | Settings $(n,m,p)$ | RSG Obj | RSG CPU | AManPG Obj | AManPG CPU | ARADMM Obj  | ARADMM CPU |
> | ------------------ | ------- | ------- | ---------- | ---------- | ----------- | ---------- |
> | (300, 20, 8)       | -3.9517 | 0.7970  | -4.2671    | 0.8555     | **-4.4071** | **0.4264** |
> | (400, 30, 10)      | -5.5595 | 1.1105  | -5.8129    | 1.0796     | **-6.0378** | **0.5383** |
> | (500, 40, 12)      | -7.1168 | 1.9584  | -7.3048    | 1.3820     | **-7.6569** | **0.6888** |
> | (600, 50, 14)      | -7.2259 | 2.5911  | -7.4608    | 1.7482     | **-7.9029** | **0.8701** |
>
> ---
>
> **Table 2: Comparison of the ALM-type methods for solving the SPCA problem with different settings**
>
> | Settings $(n,m,p,\mu)$ | Obj (ALMSSN) | CPU (ALMSSN) | Spa (ALMSSN) | Obj (ALMSRTR) | CPU (ALMSRTR) | Spa (ALMSRTR) | Obj (ARADMM) | CPU (ARADMM) | Spa (ARADMM) |
> | ---------------------- | ------------ | ------------ | ------------ | ------------- | ------------- | ------------- | ------------ | ------------ | ------------ |
> | (1500, 20, 8, 0.5)     | 6.0212       | 1.6349       | 93.10        | 4.0169        | 1.2679        | 98.38         | **3.9552**   | **0.9417**   | **99.93**    |
> | (2000, 40, 10, 0.6)    | 8.9390       | 2.7063       | 94.03        | 5.9553        | 1.9281        | 99.85         | **5.9190**   | **1.3272**   | **99.95**    |
> | (2500, 60, 12, 0.8)    | **9.4527**   | 2.9387       | 99.61        | 9.5296        | 3.6644        | 99.95         | 9.4747       | **1.8394**   | **99.96**    |
> | (3000, 80, 15, 1)      | 15.7085      | 4.7114       | 94.65        | **14.8418**   | 4.4791        | 99.96         | 14.8470      | **2.2669**   | **99.97**    |
>
>
> **Reference** （We have cited all those papers in our revised manuscript）
>
>
>
> [1] Huang, W., & Wei, K. (2023). An inexact Riemannian proximal gradient method. Computational Optimization and Applications, 85(1), 1-32.
>
> [2] Zhou, Y., Bao, C., Ding, C., & Zhu, J. (2023). A semismooth Newton based augmented Lagrangian method for nonsmooth optimization on matrix manifolds. Mathematical Programming, 201(1), 1-61.
>
> [3] Zhang, C., Xiao, R., Huang, W., & Jiang, R. (2024). Riemannian Trust Region Methods for SC 1 Minimization. Journal of Scientific Computing, 101(2), 32.

---

> > ### Comment · Reviewer_LuUY · 2025-08-04
> > **Thank you**
> >
> > Thank you for the very detailed response — it addresses all of the questions and weaknesses I raised. I also truly appreciate the authors’ efforts to supplement the experiments in such a short period of time, which I know is not easy. From the experimental results, the proposed ADMM-type algorithm not only has an advantage in terms of runtime, but also slightly outperforms ALM-type algorithms in terms of objective value, which is indeed a pleasant surprise. I am willing to raise my score.

---

> > > ### Author Response · Authors · 2025-08-04
> > >
> > > Thank you very much for your kind words and for taking the time to re-evaluate our work. We're truly grateful for your thoughtful feedback and are glad that our additional efforts helped clarify the strengths of our method.

---

### Official Review · Reviewer_xPLN · 2025-06-23

**Clarity:** 2
**Significance:** 3
**Originality:** 4
**Rating:** 4
**Confidence:** 4

**Summary:**

This paper proposes an adaptive Riemannian alternating direction method of multipliers to solve the problem of minimizing the sum of a smooth function and a non-smooth convex regularizer over a compact Riemannian submanifold embedded in Euclidean space. Unlike smoothed ADMM, the proposed method directly addresses the original nonsmooth problem. Through adaptive coordination of the step size and penalty parameters, ensuring the Lagrangian multiplier difference is bounded by the primal iterate difference, then establish an optimal iteration complexity of order $O(\epsilon^{-3})$ for finding an $\epsilon$-approximate KKT point. Numerical experiments on sparse PCA and robust subspace recovery demonstrate that ARADMM consistently outperforms state-of-the-art Riemannian ADMM variants in convergence speed and solution quality.

**Questions:**

*) In line 6, RADMM proposed by Jiaxiang Li et al. in [28] also involves only one Riemannian gradient evaluation and one proximal update per iteration.

*) In Section 1.2, the indicator function mentioned is not used in this paper and is recommended to be removed.

*) Vector transport is an important concept in manifold optimization, which is used in this paper. Its definition should be explained in Section 2.1.

*) Why does the inequality below Line 172 hold? Since the Lambda is a vector, does the inequality mean for entrywise inequality?
*) Why does the second equation in (30) hold?

*) Lemma A.1 is missing a proof or reference. Does the vector transport used in Lemma 1 need to be isometry, and if not, can the requirement of isometry in line 419 be relaxed to \|\mathcal{T}_{x}^{y}(u)\| \leq L \|u\|, where there L is some constant?

*)  Lemma B.1., B.2., and Theorem B.1. and 3.1. all require c_\rho\geq 1, but the parameter c_\rho is set to 0.6 in the numerical experiments for robust subspace recovery. Moreover, line 408 and line 420 both require c_\rho\leq 1, but line 418 needs \rho_{k-1} = c_{\rho}(k-1)^{1/3}>1 . I think the parameter c_{\rho} was chosen incorrectly and needs to be adjusted.

*)  During the proof of Theorem B1, there were several writing errors, which made reading difficult. Such as in the first inequality of the proof, \rho_{k}->\rho_{k+1}, \rho_{k-1}->\rho_{k}, (\ell_h+\lambda_{max})^2 is lost in the proof, and so on.

*)  Some symbols do not indicate their meaning before they are used. In Definition 2.1, the meaning of the symbol T\mathcal{M} is not specified. In inequality (44), the meanings of symbols f_* , h_* and \mathcal{D} are not specified. Moreover, the symbol \mathcal{D} also appears in line 142, but their meanings are different.

*)  There are many typos in this paper; for example, in line 114. It is suggested to check again.

**Ethical Concerns:**

["NO or VERY MINOR ethics concerns only"]

**Final Justification:**

I will keep my score and my questions are all answered.

**Limitations:**

This paper already states its limitation, i.e., its current analysis does not exploit structural properties like the Kurdyka–Łojasiewicz (KL) inequality, which could lead to improved convergence rates.

**Paper Formatting Concerns:**

*) It is recommended that the convergence analysis of the algorithm and the assumptions used be put together in a separate chapter.
*) This paper lacks the organization paragraph. Is it due to journal space limitations?

**Quality:**

3

**Strengths And Weaknesses:**

The idea presented in this paper is great. The innovation lies in adaptively selecting dual step sizes and penalty parameters, explicitly bounding multiplier differences by the norms of corresponding primal iterate differences. This is crucial for ensuring convergence. But there are still some details missing in the writing, mainly in the theoretical proof part. The major concern also lies in the theoretical proofs of the main result.

---

> ### Author Rebuttal · Authors · 2025-07-29
>
> Thank you for the positive feedback and for recognizing the core innovation of our work. We have addressed each concern carefully and made corresponding revisions where appropriate, as detailed below.
>
> **Reply to Questions**
> *  Thank you for the comment. To avoid confusion, we will revise the sentence to: “Our approach involves only one Riemannian gradient evaluation and one proximal update per iteration.”
> * We have removed the sentence about the indicator function.
> * We have added its definition in Section 2.1 in the revised version.
> > **Vector Transport.**
> > Given a Riemannian manifold $\mathcal{M}$, the *vector transport* $\mathcal{T}_x^y$ is an operator that transports a tangent vector $v \in T_x \mathcal{M}$ to the tangent space $T_y \mathcal{M}$, i.e., $\mathcal{T}_x^y(v) \in T_y \mathcal{M}$. In this paper, we assume $\mathcal{T}_x^y$ is isometric.
> * Thank you for pointing this out. The inequality was missing a norm, which may have led to confusion. It should be written as:
> $$
> \begin{aligned}
> \\|\lambda\_{k+1} - \lambda_k\\| &= \left\\| \beta\_{k+1} (\mathcal{A} x\_{k+1} - y\_{k+1}) \right\\| \leq \beta\_{k+1} \left\\| \mathcal{A} x_k - y\_{k+1} - \frac{\lambda_k}{\rho_k} \right\\| + \beta\_{k+1} \\| \mathcal{A}(x\_{k+1} - x_k) \\| + \beta\_{k+1} \frac{\\|\lambda_k\\|}{\rho_k} \\\\
> & \leq \beta\_{k+1} \left\\| \mathcal{A} x_k - \frac{\lambda_k}{\rho_k} - \mathrm{prox}\_{h/\rho_k} \left( \mathcal{A} x_k - \frac{\lambda_k}{\rho_k} \right) \right\\| + \beta\_{k+1} \\|\mathcal{A}\\|\_{op} \\|x\_{k+1} - x_k\\| + \beta\_{k+1} \frac{\\|\lambda_k\\|}{\rho_k}.
> \end{aligned}
> $$
> We will correct this in the revised version of the paper. Thank you again for carefully catching this issue.
> * The second equation in (30) should use "$\leq$" instead of equality. The correct derivation is as follows:
> $$
> \begin{aligned}
> \\|\beta\_{k+1} (\mathcal{A} x\_{k+1} - y\_{k+1})\\| &= \beta\_{k+1} \left\\| \mathcal{A} x_k - y\_{k+1} - \frac{\lambda_k}{\rho_k} + \mathcal{A}(x\_{k+1} - x_k) + \frac{\lambda_k}{\rho_k} \right\\|  \\\\
> & \leq \beta\_{k+1} \left\\| \mathcal{A} x_k - y\_{k+1} - \frac{\lambda_k}{\rho_k} \right\\| + \beta\_{k+1} \\| \mathcal{A}(x\_{k+1} - x_k) \\| + \beta\_{k+1} \frac{\\|\lambda_k\\|}{\rho_k}.
> \end{aligned}
> $$
> We will correct this in the revised version. Thank you again for carefully catching this oversight.
> * We thank the reviewer for the careful reading. We have added the appropriate reference for Lemma A.1, which cites Lemma 1 from \[10]. Regarding the vector transport: it is not required to be an isometry for the result in Lemma A.1 to hold. In fact, Line 419 can be relaxed to the more general condition
> $$
> \\|\mathcal{T}\_{x}^{y}(u)\\| \leq L \\|u\\|,\quad \text{for some constant } L > 0.
> $$
> In our analysis, we assumed an isometric vector transport to simplify the notation and presentation. We will clarify this assumption and the rationale for it in the revised manuscript.
> * Consistency of Assumptions on $c_\rho$.
>    - Regrading the choice of $c\_{\rho}$ in robust subspace recovery,  we have re-run the experiments with $c\_{\rho} = 1$. For ARADMM, we reset $c_\rho = 1$ and $\beta_0 = 700$, and retain the original values for $\rho_0$, $c_\tau$, and $c_\beta$.
> The implementations of other methods follow the code provided by [28] in the manuscript, where we set the ADMM stepsize $\eta = 10^{-5}$. All methods terminate when $|F(X_{k+1}) - F(X_k)| \leq 10^{-6}$ or the number of iterations exceeds 5000.
> The results in **Table 1** are averaged over 10 runs with random initialization. It is shown that the results remain consistent with those reported using $c\_{\rho} = 0.6$.
>   - We want to emphasize that the condition $c_\rho \geq 1$ is **not essential**.  The condition appears initially in Lemma A.2, where it was used to simplify the expression in the following inequality:
> $$
> \Phi_k(\mathcal{R}\_{x}(\eta)) \leq \Phi_k(x) + \langle \eta, \nabla \Phi_k(x) \rangle + \frac{\alpha^2 (\ell\_{\nabla f} + \rho_k \sigma_A^2 ) + 2 G \beta }{2} \\|\eta\\|^2.
> $$
> Assuming $c\_\rho \geq 1$ implies $\rho_k \geq 1$, which simplifies the bound to:
> $$
> \alpha^2 (\ell\_{\nabla f} + \rho\_k \sigma_A^2 ) + 2 G \beta \leq \left(\alpha^2 (\ell\_{\nabla f} + \sigma_A^2 ) + 2 G \beta\right)\rho_k = M\rho_k.
> $$
> However, this assumption is not strictly necessary. In fact, we can relax it by identifying a threshold index $\tilde{k} > 0$, such that for all $k > \tilde{k}$, we have $\rho_k \geq 1$. This does not affect the final convergence guarantees—the same convergence rate still holds. This type of strategy has been employed in prior work (e.g., [1]) to mitigate the reliance on precise problem constants. This is why $c\_{\rho} = 0.6$ also works well empirically.  In fact, the condition for other parameters can also be eliminated by this strategy, such as $\tau_k = c_\tau k^{-1/3}$.
>   - Regarding lines 408 and 420, our intention was to derive the desired bounds using conditions on $c_\tau$, but we overlooked the existence of $c_\rho$ in $\rho_k$. We will revise Equation (20) to explicitly include this dependence, i.e.,
> $$
>  c\_{\tau} \leq \frac{1}{c_{\rho}} \min\\{  \frac{1}{C}, \frac{1}{M}, \frac{1}{16 c\_{\beta}\alpha^2 \sigma_A^2}\\}.
> $$
>  With this correction, it is sufficient to assume $c_\rho \geq 1$ throughout the analysis. In addition, for line 418,  inequality should read $\rho_{k-1} = c_\rho (k-1)^{1/3} \geq 1$, not $> 1$. We will correct this in the revision.
>
> We will revise the theoretical discussion accordingly and clarify how the convergence still holds under more general parameter settings.
>
> ---
>
> ### **Table 1: Comparison of SOC, MADMM, RADMM and ARADMM for solving the robust subspace recovery problem**
>
> | p | (n, p₁, p₂)   | SOC obj  | SOC CPU | MADMM obj | MADMM CPU | RADMM obj | RADMM CPU | ARADMM obj   | ARADMM CPU |
> | - | ------------- | -------- | ------- | --------- | --------- | --------- | --------- | ------------ | ---------- |
> | 4 | (30,100,500)  | 286.5284 | 1.4835  | 286.4820  | 0.7677    | 286.4599  | 0.0486    | **286.3336** | **0.0057** |
> |   | (40,125,750)  | 363.2060 | 2.6384  | 363.1336  | 1.4823    | 363.0977  | 0.0136    | **362.8826** | **0.0119** |
> |   | (50,150,1000) | 423.6242 | 2.7054  | 423.5769  | 2.3196    | 423.5352  | 0.0378    | **423.2783** | **0.0136** |
> | 6 | (30,100,500)  | 431.2076 | 2.4063  | 431.1518  | 1.4842    | 431.1275  | 0.0160    | **431.0405** | **0.0089** |
> |   | (40,125,750)  | 542.7145 | 2.8961  | 542.6425  | 0.7155    | 542.5957  | 0.0540    | **542.4900** | **0.0112** |
> |   | (50,150,1000) | 637.8058 | 2.6239  | 637.7484  | 0.7584    | 637.6906  | 0.0972    | **637.3598** | **0.0176** |
>
>
>
> * We thank the reviewer for pointing out the typos and missing terms in the proof of Theorem B.1. We have corrected the subscript inconsistencies (e.g., $\rho_k \to \rho\_{k+1}$, $\rho\_{k-1} \to \rho_k$) and added the missing constant term $(\ell_h + \lambda\_{\max})^2$, which was inadvertently omitted in the derivation. Since this term is independent of the iteration index $k$, its omission does not affect the final complexity result. We have carefully reviewed the entire proof and corrected all related issues to improve clarity and correctness in the revised manuscript.
> * Thank you for the helpful suggestions. We have addressed the symbol-related issues as follows:
>
>   - We have now added the definition of $T\mathcal{M}$ (the tangent bundle of the manifold $\mathcal{M}$) before its first use in Definition 2.1.
>
>   - In inequality (44), the symbols $f_*$ and $h_*$ were previously undefined. We have clarified them explicitly in Assumption 3.1 as: $
>   f_* = \inf_x f(x) > -\infty, \quad h_* = \inf_x h(x) > -\infty. $
>   - Regarding the symbol $\mathcal{D}$, it is intended to denote the same quantity throughout the paper, specifically:
>   $
>   \mathcal{D} = \max_{x, y \in \mathcal{M}} \\|x - y\\|,
>   $
>   as introduced in line 142. We will explicitly reference this definition when $\mathcal{D}$ appears in inequality (44) to ensure consistency and avoid confusion.
> * We have carefully proofread the entire manuscript and corrected all identified typos and formatting issues to improve the overall clarity and presentation. We appreciate the reviewer’s suggestion and will ensure that the revised version meets a high standard of writing quality.
>
> **Reply to Limitations**
>
> Thank you for pointing this out. We acknowledge this limitation, as also noted in the paper. Our current focus is on establishing convergence without smoothing and analyzing the iteration complexity in the nonconvex, nonsmooth setting. Incorporating structural properties such as the KL inequality is indeed a promising direction and could lead to sharper rates, which we leave for future work.
>
> **Reply to Paper Formatting Concerns**
>
> We will move the assumptions to appear before the convergence analysis section. Additionally, we will add an organization paragraph to improve the clarity and guide the reader through the structure of the paper.
>
>
> [1] Lu, Z., Mei, S., & Xiao, Y. (2024). Variance-reduced first-order methods for deterministically constrained stochastic nonconvex optimization with strong convergence guarantees. arXiv preprint arXiv:2409.09906.

---

> > ### Comment · Reviewer_xPLN · 2025-08-04
> >
> > Thank the authors for the responses. I will keep the current score.

---

> > > ### Author Response · Authors · 2025-08-04
> > >
> > > Thank you for taking the time to read our response and for your thoughtful review.

---

### Official Review · Reviewer_dLzV · 2025-06-30

**Clarity:** 3
**Significance:** 2
**Originality:** 2
**Rating:** 4
**Confidence:** 3

**Summary:**

This work proposes a Riemannian ADMM algorithm for optimizing the sum of a smooth function and the composition of a non smooth function with a linear operator, all of that restricted to be in a Riemannian manifold embedded in R^n. Access to the gradient of the former function and the proximal operator of the latter is assumed. Author show that their algorithm achieves a KKT point up to an \epsilon in O(\epsilon^{-3}). Experients are provided.

**Questions:**

What are the main differences and new techniques of this work with respect to its Euclidean counterpart?

**Ethical Concerns:**

["NO or VERY MINOR ethics concerns only"]

**Final Justification:**

After the author's response, some points were clarified and I keep my evaluation unchanged. This paper falls into the borderline accept category.

**Limitations:**

yes

**Paper Formatting Concerns:**

This is minor, but the checklist should have gone after the references after the main paper and the appendix would go after the checklist.

**Quality:**

3

**Strengths And Weaknesses:**

This work is generally well written and the result seems interesting, but I missed an explanation comparing the analysis of this paper and the one of the Euclidean approach that this work is based on. There are a couple of minor problems / typos in the analysis:

The way the statement of Theorem 3.1 is phrased is inaccurate, since if the actual guarantee you prove is what you state, then the minimum of each of the three objectives could happen at different indices k, and it could be that for no single index the three terms are small at the same time. However, taking a look at your proof you can fix it and claim that you have a k such that the sum of all three terms is small.

There is an index problem in the proof of Theorem B.1. At the beginning of the proof, a property is shown for $\mathcal{L}$ with rho having an index a unit less than the indices of x, y, \lambda. However, between line 430 and 431 the indices for all of those variables become the same. This should be fixed.

One thing to note is that the method is claimed to be adaptive but the algorithm requires access to most problem constants like the smoothness of f, the operator norm of the linear map. The adaptivity refers to how the dual step size can adapt depending on the previous iterates but still depends on these constants.

---

> ### Author Rebuttal · Authors · 2025-07-29
>
> Thank you for the positive feedback and for recognizing the core innovation of our work. We have addressed each concern carefully and made corresponding revisions where appropriate, as detailed below.
>
> **1.Regarding the Comparison to the Euclidean Counterpart**:
>
> We thank the reviewer for this insightful question.  We discuss that our method differs from  Euclidean ADMM in the following two aspects:
>
> - On one hand, when the manifold reduces to $\mathbb{R}^n$, several ADMM variants with convergence guarantees have been developed (e.g., [1, 2, 3]). These methods typically rely on using the optimality condition of the $x$-subproblem and the smoothness of $f$ to derive a bound on $\\|\lambda_{k+1} - \lambda_k\\|$. However, this strategy fails to generalize to the manifold setting due to the presence of projection operators, see Section 3.1 for details. This also explains why the original Riemannian ADMM (e.g., RADMM) lacks convergence guarantees.
>
> - On the other hand, for general nonlinear constraints (including manifold constraints), some prior works [4,5,6] apply ADMM by penalizing the constraint. However, this approach does not exploit the underlying manifold geometry and essentially reduces to penalty-based methods, which are generally less efficient than manifold-aware algorithms.
>
> - In contrast, our method adopts a **fundamentally different strategy** from Euclidean ADMM to control the difference $\\|\lambda_{k+1} - \lambda_k\\|$. We achieve this through **careful design of the penalty parameter and dual step size updates**, which is of independent interest. Notably, our analysis does **not rely on the smoothness of $f$** to control $\\|\lambda_{k+1} - \lambda_k\\|$, which opens up a potential future direction: developing ADMM-type methods for problems where **both terms in the objective are nonsmooth**.
>
> We will clarify this comparison more explicitly in the revised manuscript.
>
>
> **2.Regarding the statement of Theorem 3.1**
>
> We thank the reviewer for pointing out this important issue. We agree that, as originally stated, the theorem could imply that the minimum of each of the three terms occurs at a different index $k$, which would not guarantee a single iterate where all terms are simultaneously small.
>
> It follows from (46) that:
> $$
> \sum\_{k=1}\^{K}\frac{\tau_k}{4}\left\\| \text{grad} \mathcal{L}\_{\rho_k}\left(x_\{k}, y_\{k+1}, \lambda_k\right) \right\\|^2 \leq \mathcal{G}.
> $$
> Let us denote
> $$
> R_k = \left\\| \mathcal{P}\_{T\_{x_k}\mathcal{M}} \left( -\mathcal{A}^* \bar{\lambda}_k \right) + \text{grad}f(x_k) \right\\|^2 + \text{dist}^2\left(-\bar{\lambda}_k , \partial h(y_k)\right) + \left\\| \mathcal{A}x_k - y_k \right\\|^2.
> $$
> Now we can use Lemma B.2 to bound $R_k$. Then we multiply $\tau_k$ into both sides and sum it up from $k=1$ to $K$.
> Following the proof of Theorem B.1, we can obtain the following result:
> $$
> \min\_{1 \leq k \leq K} R_k \leq \frac{\Gamma}{\sum\_{k=1}\^{K} \tau_k},
> $$
> with some constant $\Gamma$. Note that $\tau_k = c\_{\tau}k^{-1/3}$ and
> $$
> \sum\_{k=1}\^{K} k^{-1/3} \geq \sum\_{k=1}^{K} \int_k^{k+1} x^{-1 / 3} \mathrm{~d}x \geq \frac{1}{2} (K+1)^{2/3}.
> $$
> Then we have that
> $$
> \min\_{1 \leq k \leq K} R_k \leq \mathcal{O}(K\^{-2/3}).
> $$
> This guarantees the existence of a single iterate $k$ at which the sum of all three quantities is small, and thus the desired convergence rate holds at that point. We will update the theorem and corresponding discussion in the revised manuscript.
>
> **3.Regarding the proof of Theorem B.1**
>
> We thank the reviewer for carefully identifying the indexing inconsistency in the proof. At the beginning of the proof, the augmented Lagrangian should be written as:
>
> $$
> \mathcal{L}\_{\rho\_{k+1}}\left(x\_{k+1}, y\_{k+1}, \lambda\_{k+1}\right) - \mathcal{L}\_{\rho\_{k}}\left(x_k, y_k, \lambda_k\right),
> $$
>
> with $\rho$ taking indices consistent with the corresponding variables. This correction does not affect the validity of the proof or the final result.
>
> **4.Regarding the claim that our method is adaptive**
>
> Thank you for your comment. We agree that although our algorithm is described as *adaptive*, it still relies on certain problem-dependent constants.
>
> However, we would like to clarify that this dependence is **not essential**. Specifically, the algorithm involves three hyperparameters—$c\_\tau$, $c\_\rho$, and $c\_\beta$—which determine the sequences $\tau_k = c\_\tau k^{-1/3}$, $\rho_k = c\_\rho k^{1/3}$, and $\frac{c\_{\beta}}{k^{1/3} \log^2(k+1)}$, respectively.  For example, since $\tau_k = c\_\tau k^{-1/3}$, the condition on $c_\tau$ is essentially imposed to ensure that $\tau_k$ satisfies the required bound, i.e.,  $\tau_k < \min\\{  \frac{1}{C}, \frac{1}{M}, \frac{1}{16 c\_{\beta}\alpha^2 \sigma_A^2}\\}$ (Theorem 3.1).  This condition can always be satisfied after a sufficiently large number of iterations without requiring $c_\tau$. That is, for any $c_\tau$, there exists an integer $k_0 > 0$ such that for all $k > k_0$,  $\tau_k$ satisfies the condition. A similar argument applies to the other parameters. Importantly, this does not affect the final convergence guarantees—the same convergence rate still holds. This type of strategy has been employed in prior work (e.g., [7]) to mitigate the reliance on precise problem constants.
>
> Therefore, our use of the term *adaptive* was meant to emphasize that both the penalty parameter and the dual step size evolve with the iteration count $k$, rather than being fixed. A more accurate description might be that these parameters are *dynamic*, rather than *fully adaptive* in the traditional sense. We will clarify this point in the revised manuscript.
>
> **Reference**
>
> [1] Boţ, R. I., & Nguyen, D. K. (2020). The proximal alternating direction method of multipliers in the nonconvex setting: convergence analysis and rates. Mathematics of Operations Research, 45(2), 682-712.
>
> [2] Guo, K., Han, D. R., & Wu, T. T. (2017). Convergence of alternating direction method for minimizing sum of two nonconvex functions with linear constraints. International Journal of Computer Mathematics, 94(8), 1653-1669.
>
> [3] Hien, L. T. K., Phan, D. N., & Gillis, N. (2022). Inertial alternating direction method of multipliers for non-convex non-smooth optimization. Computational Optimization and Applications, 83(1), 247-285.
>
> [4] Hien, L. T. K., & Papadimitriou, D. (2024). An inertial ADMM for a class of nonconvex composite optimization with nonlinear coupling constraints. Journal of Global Optimization, 89(4), 927-948.
>
> [5] El Bourkhissi, L., Necoara, I., & Patrinos, P. (2023, December). Linearized ADMM for nonsmooth nonconvex optimization with nonlinear equality constraints. In 2023 62nd IEEE Conference on Decision and Control (CDC) (pp. 7312-7317). IEEE.
>
> [6] Hien, L. T. K., & Papadimitriou, D. (2024). Multiblock ADMM for nonsmooth nonconvex optimization with nonlinear coupling constraints. Optimization, 1-26.
>
> [7] Lu, Z., Mei, S., & Xiao, Y. (2024). Variance-reduced first-order methods for deterministically constrained stochastic nonconvex optimization with strong convergence guarantees. arXiv preprint arXiv:2409.09906.

---

> > ### Comment · Reviewer_dLzV · 2025-08-03
> > **reply**
> >
> > thank you for your reply, I'll keep my score

---

> > > ### Author Response · Authors · 2025-08-04
> > >
> > > Thank you for taking the time to read our response and for your thoughtful review.

---

### Official Review · Reviewer_prAa · 2025-07-04

**Clarity:** 3
**Significance:** 3
**Originality:** 3
**Rating:** 5
**Confidence:** 3

**Summary:**

The authors study nonsmooth nonconvex optimization over Riemannian Submanifolds. Canonical examples of such a problem include sparse and/or nonnegative PCA. An adaptive ADMM algorithm is proposed which notably does not necessitate smoothing the nonsmooth term. By carefully adapting the dual step size, ARADMM controls the bound on the dual variables and the difference between successive dual variables. This yields an optimal oracle / iteration complexity of O(e-3) (under certain flexible assumptions e.g. Lipschitzness and smoothness of the smooth term and convexity and Lipschitzness of the nonsmooth term). Compared to existing ADMM-based frameworks for nonsmooth optimization over Riemannian Manifolds (e.g. MADMM), the proposed method only requires a single projection (via the retraction defined on M) and avoids nested iterative procedures or complicated subroutines (in addition to not modifying the original problem, e.g. by smoothing).

Numerically, the authors compare RADMM to four ADMM-based methods and two first-order nonsmooth manifold optimization algorithms. On Sparse PCA, RADMM achieves a marked improvement over other methods in terms of solution quality and empirical convergence rate. On a robust subspace recovery task formulated via dual principal component pursuit, RADMM also outperforms relevant ADMM-based methods and a recently proposed energy minimization scheme.

**Questions:**

See weaknesses. I would like some contextualization of your framework with respect to some existing relevant missing references and more experimental results if possible.

**Ethical Concerns:**

["NO or VERY MINOR ethics concerns only"]

**Final Justification:**

The authors response to my rebuttal is comprehensive and detailed. I increase my confidence in acceptance.

**Limitations:**

yes

**Quality:**

3

**Strengths And Weaknesses:**

**Strengths** This is a well-written paper that contributes a new practically designed and provably convergent ADMM-based algorithm for nonsmooth nonconvex optimization on smooth manifolds. The derived method notably does not necessitate smoothing the nonsmooth terms in the objective while maintaining numerical efficacy.

Numerically, the method shines over relevant ADMM and first-order methods for optimization over smooth manifolds in terms of simplicity of implementation, solution quality, iteration complexity, and runtime.


**Weaknesses** I am admittedly not up to date with the literature, but it seems that the author is missing some key references, including Wang, Yin, Zheng Global Convergence of ADMM in Nonconvex Nonsmooth Optimization, which also addresses nonsmooth nonconvex optimization of an ADMM-based method. It would be good to contextualize the work the authors have done with this important paper. Similarly, only two applications are discussed. It would strengthen the paper to include more experiments - e.g. nonnegative PCA as discussed in the introduction, or problems on manifolds other than the Stiefel manifold.

---

> ### Author Rebuttal · Authors · 2025-07-29
>
> We sincerely thank the reviewer for the positive feedback on the clarity of the writing and the contributions of our work. We are glad that the theoretical and practical aspects of the proposed ADMM-based algorithm were found valuable. We have addressed each concern carefully and made corresponding revisions where appropriate, as detailed below.
> * Our current version mainly focuses on summarizing existing Riemannian ADMM approaches. In the revised version, we will include a more comprehensive discussion of Euclidean-space ADMM methods for nonsmooth nonconvex problems.  We will discuss how our method differs from  Euclidean ADMM in the following two aspects:
>
>   - On one hand, when the manifold reduces to $\mathbb{R}^n$, several ADMM variants with convergence guarantees have been developed (e.g., [1, 2, 3] and the mentioned reference). These methods typically rely on using the optimality condition of the $x$-subproblem and the smoothness of $f$ to derive a bound on $\\|\lambda_{k+1} - \lambda_k\\|$. However, this strategy fails to generalize to the manifold setting due to the presence of projection operators, see Section 3.1 for details. This also explains why the original Riemannian ADMM (e.g., RADMM) lacks convergence guarantees.
>
>   - On the other hand, for general nonlinear constraints (including manifold constraints), some prior works [4,5,6] apply ADMM by penalizing the constraint. However, this approach does not exploit the underlying manifold geometry and essentially reduces to penalty-based methods, which are generally less efficient than manifold-aware algorithms (e.g., Riemannian gradient and retraction).
>
> * We appreciate the suggestion of adding an experiment on nonnegative PCA. However, we would like to clarify that the nonsmooth term in nonnegative PCA is an indicator function of the nonnegative orthant, which is not Lipschitz continuous. This violates Assumption 3.1 in our paper, which requires the nonsmooth component $h$ to be Lipschitz. For this reason, our algorithm and theoretical analysis do not directly apply to this setting, and we are unable to include it as a valid benchmark in our experiments.
>
> * To further demonstrate the applicability of our algorithm to broader settings, we have added a new experiment: Regularized Linear Classification over the Sphere \[7,8]. Consider a classification problem with $N$ training examples $\{a_i,b_i\}_{i=1}^N$ where $a_i \in \mathbb{R}^{m \times 1}$ and $b_i \in \\{-1,1\\}$ for all $i \in [N]$. The goal is to estimate a linear classifier parameter on the sphere manifold, $x \in \mathbb{S}^{m-1} := \{x \in \mathbb{R}^m : x^\top x = 1\}$, that minimizes a smooth nonconvex loss function with an $\ell_1$-regularization:
>
> $$
> \tag{1}
> \min_{x \in \mathbb{S}^{m-1}} \sum_{i=1}^{N} \left(1 - \frac{1}{1 + \exp(-b_i x^\top a_i)} \right)^2 + \mu \\|x\\|_1.
> $$
>
> For data generation, the true parameter $x$ is sampled from $\mathcal{N}(0, I_m)$ and projected onto $\mathbb{S}^{m-1}$. The features $\\{a_i\\}\_{i=1}^N$ are sampled independently, and the labels $b_i$ are set to 1 if $x^\top a_i + \epsilon_i > 0$, where noise $\epsilon_i \sim \mathcal{N}(0, \sigma^2)$, and -1 otherwise. All algorithms use the identical random initialization and terminate when $|F(X_{k+1}) - F(X_k)| \leq 10^{-8}$ or after 500 iterations, with $\mu = 0.2$ fixed in (1).
>
> For the three fixed methods, we set the penalty parameter $\rho = 150$ and the step size $\eta = 0.01$ for SOC, MADMM, and RADMM. For the two adaptive methods, we set $\beta_0 = \rho_0 = c_\beta = 100$, $c_\rho = 1$, and $c_\tau = 0.05$ for ARADMM, and use the same settings as in our SPCA experiment for OADMM. Here, the choices of parameters for all the algorithms in the experiment are consistent with the corresponding theories. Before presenting the numerical comparisons, we remind the reader that there is no convergence guarantee for SOC and MADMM.
>
> Table 1 reports the function value (denoted by “Obj”) and CPU time (in seconds), where the results are averaged over 10 repeated experiments with random initializations. From Table 1, we can see that ARADMM and MADMM quickly decrease the objective value, whereas both ARADMM and SOC achieve a lower objective value for the outputs. Moreover, ARADMM is more advantageous than existing methods in more challenging scenarios. In short, ARADMM is more efficient in terms of CPU time and objective value for test instances.
>
> ---
>
> **Table 1: Comparison of the ADMM-type methods for solving problem (1) with different settings. The best and second-best results are marked in bold and italics, respectively.**
>
> | Settings $(m,n,\sigma^2)$ | SOC Obj    | SOC CPU | MADMM Obj | MADMM CPU | RADMM Obj | RADMM CPU | OADMM Obj | OADMM CPU | ARADMM Obj | ARADMM CPU |
> | ------------------------- | ---------- | ------- | --------- | --------- | --------- | --------- | --------- | --------- | ---------- | ---------- |
> | (200, 1000, 1)            | **0.7004** | 1.1009  | 0.7340    | *0.0729*  | 0.7340    | 0.0876    | 0.7282    | 0.1024    | *0.7370*   | **0.0507** |
> | (400, 5000, 5)            | *0.6877*   | 2.2876  | 0.8267    | *0.1701*  | 0.8267    | 0.1864    | 0.7288    | 0.2799    | **0.6875** | **0.1073** |
> | (600, 10000, 10)          | *0.6469*   | 7.8231  | 0.9216    | *0.6692*  | 0.9216    | 0.7153    | 0.6665    | 1.0664    | **0.6464** | **0.4197** |
> | (800, 20000, 50)          | *0.6606*   | 30.7367 | 1.0398    | *2.2673*  | 1.0396    | 2.7062    | 0.6871    | 2.7062    | **0.6602** | **1.4167** |
>
>
>
> [1] Boţ, R. I., \& Nguyen, D. K. (2020). The proximal alternating direction method of multipliers in the nonconvex setting: convergence analysis and rates. Mathematics of Operations Research, 45(2), 682-712.
>
> [2] Guo, K., Han, D. R., \& Wu, T. T. (2017). Convergence of alternating direction method for minimizing the sum of two nonconvex functions with linear constraints. International Journal of Computer Mathematics, 94(8), 1653-1669.
>
> [3] Hien, L. T. K., Phan, D. N., \& Gillis, N. (2022). Inertial alternating direction method of multipliers for non-convex non-smooth optimization. Computational Optimization and Applications, 83(1), 247-285.
>
> [4] Hien, L. T. K., & Papadimitriou, D. (2024). An inertial ADMM for a class of nonconvex composite optimization with nonlinear coupling constraints. Journal of Global Optimization, 89(4), 927-948.
>
> [5] El Bourkhissi, L., Necoara, I., & Patrinos, P. (2023, December). Linearized ADMM for nonsmooth nonconvex optimization with nonlinear equality constraints. In 2023 62nd IEEE Conference on Decision and Control (CDC) (pp. 7312-7317). IEEE.
>
> [6] Hien, L. T. K., & Papadimitriou, D. (2024). Multiblock ADMM for nonsmooth nonconvex optimization with nonlinear coupling constraints. Optimization, 1-26.
>
> [7] Zhao, L.; Mammadov, M.; and Yearwood, J. 2010. From convex to nonconvex: a loss function analysis for binary classification. In 2010 IEEE international conference on data mining workshops, 1281–1288. IEEE
>
> [8] Zhang, D.; and Davanloo Tajbakhsh, S. 2023. Riemannian stochastic variance-reduced cubic regularized Newton method for submanifold optimization. Journal of Optimization Theory and Applications, 196(1): 324–361.

---

> > ### Comment · Area_Chair_A678 · 2025-08-05
> >
> > Dear Reviewer,
> >
> > To facilitate further evaluation, please respond to the authors’ rebuttal.
> >
> > AC

---

> ### Comment · Reviewer_prAa · 2025-08-05
>
> I appreciate the reviewer's detailed and thoughtful reply to my review. Particularly the clarifications regarding nonsmooth / nonconvex ADMM algorithms. It makes me more confident about the contribution and I maintain my support for this paper's acceptance by raising my confidence.

---

> > ### Author Response · Authors · 2025-08-05
> >
> > We greatly appreciate your thoughtful comments and are glad that our clarifications regarding nonsmooth/nonconvex ADMM were helpful. Thank you for your support and for raising your score.

---

### Note · Authors · 2025-08-13

- We thank the reviewers for their positive feedback. They recognized that our paper is well-written, logically structured, and presents a novel ADMM-based algorithm for nonsmooth, nonconvex optimization on smooth manifolds with provable convergence guarantees. The method achieves the convergence complexity of smoothing-based approaches without requiring smoothing.

- We have thoroughly addressed all raised concerns — including a comprehensive literature review on Euclidean ADMM, parameter dependencies in our algorithm, the inclusion of additional baselines,  and other experimental clarifications. Reviewers expressed satisfaction with these revisions.

---

### Decision · Program_Chairs · 2025-09-17

**Decision:**

Accept (poster)

**Comment:**

This paper studies the problem of minimizing the sum of a smooth function and a nonsmooth convex regularizer over a compact Riemannian submanifold embedded in Euclidean space. The authors proposed an adaptive Riemannian ADMM that requires only one gradient evaluation and one proximal update per iteration, attaining optimal iteration complexity. All reviewers give positive scores and highlight both the novelty and practical relevance of the work. I recommend acceptance.